# The deglacial forest conundrum

Anne Dallmeyer [1] ✉, Thomas Kleinen [1], Martin Claussen [1,2], Nils Weitzel [3,4], Xianyong Cao[5] & Ulrike Herzschuh [6,7,8]

How fast the Northern Hemisphere (NH) forest biome tracks strongly warming climates is largely unknown. Regional studies reveal lags between decades and millennia. Here we report a conundrum: Deglacial forest expansion in the NH extra-tropics occurs approximately 4000 years earlier in a transient MPI-ESM1.2 simulation than shown by pollen-based biome reconstructions. Shortcomings in the model and the reconstructions could both contribute to this mismatch, leaving the underlying causes unresolved. The simulated vegetation responds within decades to simulated climate changes, which agree with pollen-independent reconstructions. Thus, we can exclude climate biases as main driver for differences. Instead, the mismatch points at a multi-millennial disequilibrium of the NH forest biome to the climate signal. Therefore, the evaluation of time-slice simulations in strongly changing climates with pollen records should be critically reassessed. Our results imply that NH forests may be responding much slower to ongoing climate changes than Earth System Models predict.

Pollen records are the most widespread terrestrial archives for climate and vegetation reconstructions[1,2]. They have been commonly used to evaluate simulations based on Earth System Models (ESMs), mainly for time-slices in the recent geological past, such as the mid-Holocene (6 ka, i.e., 6000 years before present)[3,4] or the Last Glacial Maximum (LGM, approximately 21,000 years before present)[5]. Long-term transient ESM simulations including vegetation dynamics have rarely been performed so far, and a large-scale, synchronised overview of pollen-based vegetation reconstructions was previously only available for very few time slices[6]. However, it is not possible to assess to what extent reconstructed and simulated vegetation dynamics deviate when looking at time slices alone. Most importantly, it has not yet been possible to analyse to what extent the reconstructed hemispheric-wide vegetation distribution is in equilibrium with the climatic conditions prevailing at the reconstructed time, since pollen-independent climate reconstructions are rare. Therefore, it cannot be predicted how fast forests will respond to the expected rapid temperature and hydroclimate changes

in the future[7,8] and if trees can migrate fast enough to track their climate niche or adapt to the projected climate changes. If disequilibrium predictions are correct, many plants could become extinct[9]. This is a major source of uncertainty in the prediction of how climate change affects biodiversity[10] and biogeographic dynamics.

During the LGM, the global climate was much drier and colder than today[11,12], and most of the Northern Hemisphere (NH) extratropics were characterised by open landscapes[13]. Trees survived in refugia[14,15], but their locations and the geography and timing of the re-colonisation by trees are debated. Previously suggested southern refugia would have to result in high dispersal rates and would require rapid migration of tree species during the deglaciation, e.g., by infrequent long-distance dispersal events[16].

Most plant species have changed their geographical range since the LGM, some of them thousands of kilometres[17]. The rate of migration is species-dependent and varies in space and time. Studies on fossil pollen records reveal both, fast (i.e., closely following the climate

[1]Max Planck Institute for Meteorology, Bundesstrasse 53, 20146 Hamburg, Germany. [2]Centrum für Erdsystemforschung und Nachhaltigkeit (CEN), Universität Hamburg, Bundesstrasse 55, 20146 Hamburg, Germany. [3]Institute of Environmental Physics, Heidelberg University, Im Neuenheimer Feld 229, 69120 Heidelberg, Germany. [4]Department of Geosciences, University of Tübingen, Schnarrenbergstr. 94-96, 72076 Tübingen, Germany. [5]Alpine Paleoecology and Human Adaptation Group (ALPHA), State Key Laboratory of Tibetan Plateau Earth System, and Resources and Environment (TPESRE), Institute of Tibetan Plateau Research, Chinese Academy of Sciences, Beijing, China. [6]Alfred Wegner Institute, Helmholtz Centre for Polar and Marine Research, Potsdam, Germany. [7]Institute of Environmental Sciences and Geography, University of Potsdam, Potsdam, Germany. [8]Institute of Biochemistry and Biology, University of Potsdam, Potsdam, Germany. ✉e-mail: anne.dallmeyer@mpimet.mpg.de

change) and slow (i.e., lagging climate forcing) vegetation responses to the deglacial climate change[18]. For instance, some marine and terrestrial records reveal a vegetation dynamic in line with the millennial climate variability in parts of the northern extratropics[19], in particular along the North Atlantic[20]. However, these studies do not include records from the Asian continent. Fast responses have also been reported for some tree species in Europe[21] and woody taxa in North America[22]. In contrast, most tree types in Europe are still responding to the deglacial climate signal and have not yet reached their potential geographical range[23]. Similarly, the geographical range of major tree species in eastern North America is not in equilibrium with climate[24]. Thus, it is not yet clear, how well the forest biome in a hemispheric perspective tracked the climate changes since the LGM and how the hemispheric forest will respond to future climate changes.

Climate is seen as the main determinant of geographical ranges at broad spatial scales[25]. However, many ecological processes act on longer timescales than the rate of strong climate changes and could thereby induce a lag in the vegetation response to the climate forcing[18]. In particular, colonisation of previous unvegetated areas or bare rocks can be slow[17]. (Re-)colonisation depends not only on dispersal rates but also on soil conditions, population buildup and size, interspecies competition and evolutionary adaption of the plants[8,18]. Complicating this, pollen data are in some aspects limited in their ability to record vegetation dynamics[26], although much effort is done to overcome these caveats[2]. For instance, pollen productivity can be strongly decreased under lower $CO_2$ levels and temperatures[27] and the coupling of vegetation to environmental factors is complicated, challenging the reliability of pollen-based vegetation reconstructions for glacial climates.

Here we present a transient simulation of the last 22000 years conducted with a comprehensive Earth System Model, the MPI-ESM 1.2, with dynamically coupled vegetation (see Methods). We compare the simulated forest cover change with the recently published synthesis of biome reconstructions for the NH[28] to challenge our understanding of vegetation dynamics in a globally warming climate.

## Results and discussion
### Simulated climate and forest cover change

The simulated changes in near surface air temperature closely resemble reconstructions on hemispheric and regional levels (Fig. 1, Supplementary Fig. 1, Supplementary Table 2) and are mostly in line with a previous, widely used simulation of the deglaciation (TraCE-21 ka)[29]. NH temperatures increase slowly until the Bølling/Allerød period (BA, 14.69-12.9 ka) and drop abruptly into, and rise again out of, the Younger Dryas (YD, 12.9–11.6 ka), respectively (Fig. 1b). During the Holocene, NH temperature increases again, albeit at a slower rate than the more recent reconstruction by Osman et al.[30] indicates. The simulated amplitude of the LGM to mid-Holocene temperature rise is approximately 1 K larger compared to Shakun et al.[31], but substantially smaller than in the Osman et al.[30] reconstruction (approximately 3 K). However, the simulated difference in global mean temperature between LGM and late-Holocene temperature of about 5 K is in agreement with a most likely range of 5–7 K by recent state estimates[12,30].

The regional temperature changes follow the NH dynamics (Supplementary Fig. 1). The amplitude of the simulated YD temperature drop is similar to the GISP2-based temperature reconstruction[32]. The simulation agrees best with reconstructions for Europe and North America, but the amplitude in the LGM to Holocene temperature change, as well as the multi-millennial variability in Asia, seems to be underestimated by the model (Supplementary Fig. 1b, c).

Vegetation independent reconstructions for precipitation are rare and mostly available in the form of oxygen isotope records whose interpretation with respect to their ability to record rainfall is debated[33]. Overall, our model appears to lead to a rather too dry LGM and a stronger increase in precipitation during the deglaciation than expected by reconstructions (Fig. 1c, e). The largest deviation from the

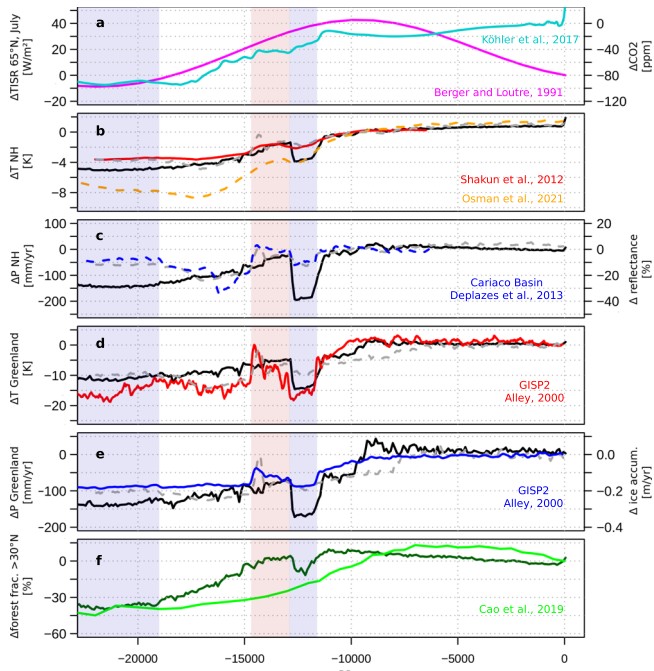

**Fig. 1 | Mean climate and forest cover change in the Northern Hemisphere.** Temperature and precipitation reconstructions for the last 22000 years before present (BP) compared to the transient MPI-ESM 1.2 simulation (black) and TraCE-21ka simulation[29] (grey). **a** July insolation anomaly to 0 ka (ΔTISR) at 65°N[78] (magenta) and $CO_2$ forcing anomaly to pre-industrial[67] (turquoise) as used as forcing in this simulation. **b** Northern Hemisphere (NH) temperature anomaly reconstructed by Osman et al.[30] (orange) and Shakun et al.[31] (red). **c** simulated NH precipitation change vs. anomaly to Holocene mean (11.5–6.5 ka) in total reflectance in Cariaco basin sediments[79] (blue), assuming that tropical precipitation largely controls the overall NH dynamics. **d** GISP2 Greenland temperature anomaly[32] to pre-industrial (PI) (red). **e** GISP2 Greenland ice accumulation[32] (blue) vs simulated Greenland precipitation as anomaly to pre-industrial. **f** Reconstructed[28] (light green) and simulated (dark green) NH forest biome cover fraction as difference to 0 ka. The last glacial maximum period (22–19 ka, light blue shading), Bølling/Allerød (14.6–12.9 ka, light red shading) and the Younger Dryas period (12.9–11.6 ka, light blue shading) are marked.

reconstructed moisture change occurs in the northern part of the East Asian summer monsoon for which the model calculates slightly increasing annual mean precipitation since the mid-Holocene (Supplementary Fig. 1g).

The simulated amplitude of change in the forest biome cover fraction north of 30°N between LGM and mid-Holocene agrees well with the reconstructed amplitude of forest cover change presented by Cao et al.[28] (Fig. 1f). However, the onset of deglacial forest expansion occurs much earlier in the model than in the reconstructions. In addition, the turnover rate from open glacial landscapes to dense Holocene forest cover is much faster in the model, leading to an Early-Holocene forest maximum (around 11 ka) in the model, while the reconstructed forest distribution peaks during the mid-Holocene (around 7 ka). This "deglacial forest conundrum", i.e., the temporal mismatch of 4000 years in the deglacial increase of the NH forest cover, may be an indication that the natural vegetation at hemispheric scale is out of equilibrium with the prevailing climate on Earth on multi-millennial timescales. This is much longer than the implemented timescales of vegetation adjustment to the climate in ESMs (decades or centuries at most)[34].

### Detailed comparison reveals largest mismatch for Asia
The synthesis of biome reconstructions generally faces the problem of regional and temporal imbalances in the record density (Supplementary

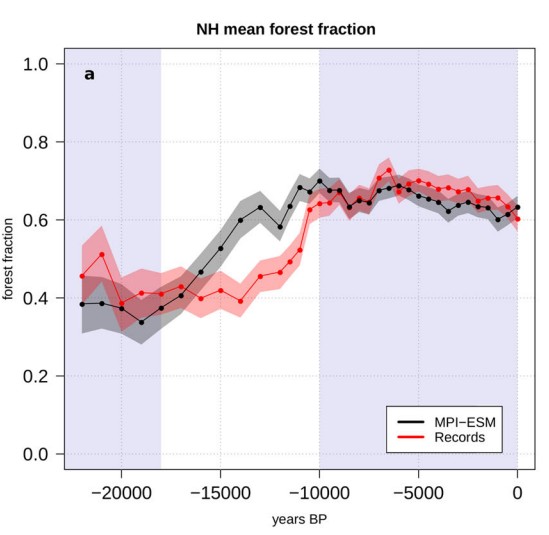
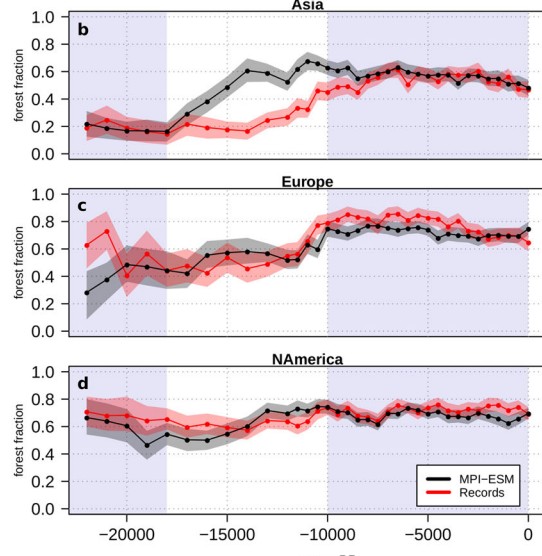

**Fig. 2 | Change in NH forest coverage.** Simulated (black) and reconstructed (red) forest cover and its uncertainty (black and red shadings) on the Northern Hemisphere (NH) and for individual continents, based on grid-cells. For this, the record and model samples have been aggregated into a regular longitude-latitude grid (10° x 5°) and averaged in these grid-cells (see Methods). Afterwards the grid-cells have been averaged: the mean cover fraction is shown for **a** NH > 30°N; **b** Asia; **c** Europe; and **d** North America. The late-glacial period (22-18 ka) and the Holocene (10-0 ka) are shaded in blue.

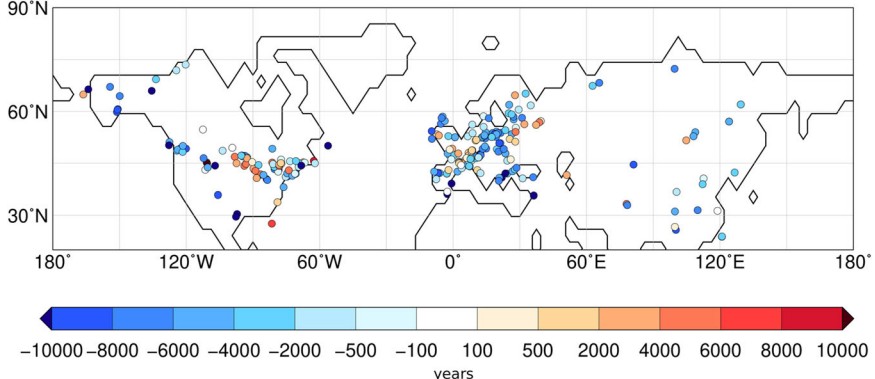

**Fig. 3 | Difference in simulated and reconstructed timing of the minimum in openness.** Difference in the timing of the minimum in the landscape openness between the model and the reconstructions for each site. For the reconstructions, the minimum in openness has been calculated on the basis of the biome affinity scores (Methods). For this purpose, the highest affinity score within the non-forest biomes has been subtracted from the highest affinity score within the forest biomes, reflecting the degree in openness. When this difference is largest, the openness is at its minimum and the plant functional type (PFT) sample is most similar to forest. Likewise for the simulation, the openness has been calculated by subtracting the total cover fraction of all non-forest PFTs from the total cover fraction of all forest PFTs. Only sites with long-lasting records have been considered. Blue means, the minimum in openness is reached earlier in the model than in the reconstructions; red means, the simulated minimum in openness lags the reconstructed minimum. This map has been prepared with the software GMT, version 5.4.3[80].

Fig. 2, "Methods"). Therefore, we only consider model grid-cells and time steps for which reconstructions are available and aggregate both on a regular longitude-latitude grid in the following (see Methods).

During the late-glacial period (22–18 ka), the reconstructions indicate more forest cover north of 30 °N than the model, which is mainly related to a difference between simulated and reconstructed forest extent in Europe and North America (Fig. 2c, d). The mean forest fraction for Asia is well reproduced by the model (Fig. 2b) for this period. The strong and continuous increase in simulated NH forest cover beginning at 18 ka is mainly caused by the expansion of the Asian forests in the model. In the reconstructions, the mean NH forest extent stays relatively constant until 14 ka. Afterwards, the Asian forest starts to increase but considerably slower than in the model. At 11 ka, the reconstructed NH forest cover rises sharply which is related to a jump from approximately 55% to nearly 80% forest coverage in Europe. The

temporal mismatch in the Asian forest dynamic continues into the Holocene until the reconstructed forest cover peaks at around 6.5 ka.

A closer look into the spatial pattern reveals locally varying leads and lags between the simulated and reconstructed forest cover change (Fig. 3). This is mainly an imprint of the heterogeneous signal in the reconstructions, also within regions. Averaged over the entire last 22000 years, the regions with the largest deviation in forest cover are the moisture-limited transition zones of temperate forest to steppe in Central North America and East Asia, in which the model constantly shows a broader extension of the forest compared to the records, and Siberia (Fig. 4). Despite regional model-data mismatches in North America and Europe, the simulated large-scale dynamics on both continents are mostly consistent with the reconstructions. In contrast, the reconstructions show a consistently later deglacial forest increase in Asia. Therefore, the deglacial forest conundrum can mainly be

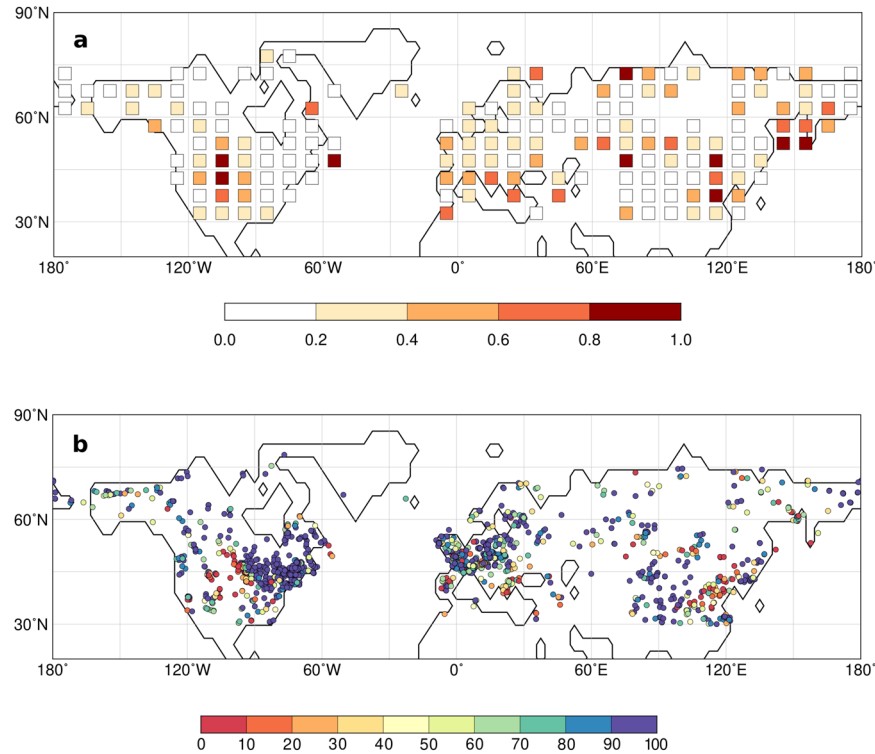

**Fig. 4 | Agreement between the simulated and reconstructed forest distribution.** The simulated forest distribution at each time-step is compared to the reconstructed forest. **a** shows the mean difference in grid-cell forest cover for the entire period (22 kyrs), i.e., the sum of differences in forest cover in each grid-cell over all time-steps divided by the available time-steps. **b** shows the percent agreement between the simulated and reconstructed vegetation cover (forest vs. open land) at each site for all time-steps. The maps have been prepared with the software GMT, version 5.4.3[80].

attributed to the mismatch between simulation and reconstruction at Asian sites (mainly Siberia and the East Asian monsoon region).

## Temporal differences in NH forest expansion related to delayed vegetation response to climate signal?

The conundrum could be caused by (a) shortcomings in biomisation techniques and/or insufficient raw data (e.g., the low record density in Asia), (b) oversimplified vegetation and absence of soil dynamics in ESMs or (c) biases in the simulated climate.

Potential causes (a): The pollen-based biomisation method is validated by comparisons between reconstructed biomes and modern vegetation distributions[35,36]. However, fossil pollen assemblages may lack modern analogues for example in heavily human-altered landscapes[37] or because past climate and atmospheric compositions differ from modern conditions[38]. This may lead to less reliable taxa-to-PFT and PFT-to-biome assignments, that furthermore depend at least partly on subjective choices[39]. Similarly, the model-based biomisation technique relies on only few PFTs and constant cover fraction limits and bioclimatic thresholds that have been defined from modern observations and may thus not be valid for strongly deviating climates. The PFT-biomisation procedure is rather simple. It competes with other model biomisation techniques, but has—similar to these methods—difficulties to present biomes that are highly controlled by non-climatic parameters (such as the savanna) or multi-faceted biomes such as tundra[36] (Supplementary Fig. 5). Tundras can appear as a very open landscape with predominantly herbaceous vegetation, but can also contain a substantial proportion of small and shrubby trees (e.g., *Betula, Alnus* and *Salix*). Complicating this, taxa of tundra often have multiple life forms (e.g., shrub or tree) and a very wide ecological range. These problems make it challenging to clearly distinguish tundra from boreal forest, also in the reconstructions. With the limited number of PFTs in the vegetation models and the strict and simple

definition of tundra in the simulated PFT-biomisation scheme, these (intra-biome) peculiarities cannot be taken into account. The differing definitions of the tundra biome in the model world and for the reconstructions may contribute to the mismatch in forest dynamic.

Pollen can be transported over long distances, e.g., from forested areas to non-forested areas, leading to inadequate separation of ecotones[40], affecting in turn the biomisation results. Different taxa have different pollen productivities that lead to systematic misrepresentations in the pollen assemblages[41]. For instance, larch, the main tree genus in Eastern Siberian boreal forests, is a poor pollen producer, and forest cover is constantly underestimated in reconstructions[42]. The error is addressed by weighting larch pollen percentage when calculating biome scores[41], but uncertainties about how dense and large the forests have been in the past remain. In contrast, pine is a strong pollen producer and expands relatively late in Asia[41] and may thus bias the mean forest cover change.

Particularly for Asia, the amount of high-quality pollen records are low. Hardly any sites exist in the central parts of Eastern Siberia and the records are limited in the Russian Far East (Supplementary Fig. 2). The low data coverage may also imply that the biomisation methods are less well calibrated in this region. Although the aggregation of pollen sites into grid cells partly overcomes the problem of imbalanced record densities, the poor data coverage in Asia may affect the calculated mean forest cover change. Since Siberian vegetation is more homogenous than on other continents and mega-biome belts are large in extent, the few records can nevertheless be representative at regional-to-continental scale[43].

One possibility to overcome the problems of the biomisation methods would be the direct comparison of the simulated tree PFTs with pollen-based, quantitative estimates of tree coverage such as provided by the REVEALS-model[44]. In this respect, much progress has been made in recent years and datasets, mostly for the last 10,000

years, have been published[41,45]. However, a dataset covering the entire NH and also the deglaciation period is not available yet and some problems, e.g., regarding temporal changes in pollen productivity through this period also affect REVEALS results.

Potential causes (b): Most Dynamic Global Vegetation Models (DGVMs) implemented in ESMs, including the MPI-ESM, assume a rapid response of plant functional types (PFTs) to climate, typically within decades or centuries. Furthermore, seeds for all plants are supposed to be available everywhere[46] and suitable areas for plant establishment mostly depend on bioclimatic thresholds. The spatial resolution of the simulation used here is coarse, leading to an underestimation of the orography which may, in reality, act as barrier for seed dispersal. Plant dispersal characteristics largely determine the ability of taxa to track climate change[47]. Limited dispersal may lead to multi-millennial migrational lags as proposed e.g., for some European tree taxa[48], but also for Siberian forest taxa[49,50]. Temperature reconstructions for northern Europe from macrofossils reveal a much warmer climate than pollen-based reconstructions until tree establishment is mostly completed, pointing to an effect of tree migration on climate reconstructions[51].

For Siberia, misadaptation of the trees to the local glacial climate conditions in the refugia is discussed[52], which probably has led to genetically exhausted and non-competitive populations and which would have necessitated a reinvasion of these taxa from southern populations[50]. This could explain the slow response in the reconstructions compared to the fast response in the model lacking a dispersal routine.

Among other factors (for an overview see Williams et al.[18]), the expansion of plant species depends on their population growth rate and population size, local evolutionary adaption[53], interspecific competition[54] or high dependencies on other species. These mechanisms are simplified or even not considered in ESMs. Tree species with long generation times may be less competitive compared to resident vegetation that has adapted or acclimatised to unsuitable climatic conditions[53], whereas tree PFTs are commonly favoured over grass in MPI-ESM[34]. Therefore, the Siberian steppe-tundra ecosystem that persisted during the LGM may be more stable than ecosystems that develop freely on periglacial ground in the deglaciated areas of Europe or North America[49].

Establishment of trees may also be hampered by unsuitable soil conditions. The buildup of soils on periglacial areas requires at least several centuries[55]. How long the primary succession lasts in total is unknown. Resident vegetation may change the soil and thereby hinder the establishment of other plants that are not able to live on these soils[56] or competition may hinder the establishment of tree seedlings. Large parts of Siberia are underlain with permafrost that may hamper the establishment of e.g., pines that require deep grounds[49]. Neither a soil development module nor a permafrost module is currently implemented in most ESMs.

How strong postglacial migration lags are, is highly debated. The rate of migration depends on the species and varies in space and time. Many studies suggest no lags[19,20] or relatively short lags of 1500 years at most[26,57]. Others state that many plant species have not yet reached equilibrium with climate in the current interglacial[18], particularly the Siberian larch forest[49] and European trees[10,23]. However, it remains questionable whether the lags observed at species level are transferable to the continental-scale forest distribution and whether the conclusions drawn from European vegetation can be transferred to other regions. Most ESMs, including the MPI-ESM, distinguish only few PFTs, thereby oversimplifying plant diversity. Mismatches in simulated forest cover may be amplified by vegetation-atmosphere interactions that may contribute to the more rapid forest expansion in the model compared to the reconstructions, at least on regional level.

Potential causes (c): The simulated climate change can be affected by systematic climate biases. The vegetation distribution in the East

Asian summer monsoon region is mainly controlled by the available moisture[58]. Comparison of simulated and observed modern climate reveals a too wet climate in East Asia (Supplementary Fig. 3), but the overall temporal evolution in precipitation during the deglaciation is in agreement with oxygen isotope reconstructions (Supplementary Fig. 1g, h). This could indicate a systematic overestimation of precipitation in the East Asian forest-steppe-transition zone and may explain the systematic overestimation of the simulated forest cover in that region. However, pollen-independent moisture reconstructions indicate that in large parts of East Asia the wettest conditions have already been achieved in the early Holocene and tend to indicate earlier wetter conditions than pollen-based reconstructions do (Supplementary Fig. 4).

Furthermore, potential refugia for temperate trees during the LGM are found in East Asia[59]. Thus, the establishment of trees early during the deglaciation is climatically not implausible. However, these populations would be small with low pollen and seed production. They may neither contribute to the migration of trees nor be detectable in the pollen records. Studies on macrofossils reveal that scattered populations of tree species can remain undetected by pollen data for several thousand years (cf. Jackson et al.[17] and references therein).

For Siberia, warm season temperature may substantially drive forest changes, as their temporal evolutions are in phase in the simulation, but precipitation may also be important[28]. The comparison with modern observations shows a tendency towards too cold climate during summer (Supplementary Fig. 3a). Pollen-independent climate reconstructions for the last 22,000 years are rare for this region. Therefore it is unclear, how warm the climate and how dense the forest during the BA has been and whether both are overestimated by the model. The simulated general NH temperature change is in line with reconstructions (Fig. 1b). We compare the simulated temperature of the warmest month with chironomid-based reconstructions for the last 12000 years[60]. The reconstructions reveal a similar, albeit weaker, dynamic as simulated, clearly pointing at an Early Holocene temperature maximum around 10.5 ka (Fig. 5c). The simulated change in openness at sites with chironomid-based reconstructions is in phase with the changes in temperature, showing a minimum in openness (i.e., maximum in forest extent) during the period of highest temperatures in the Holocene around 10.5 ka and an increase in openness afterwards, consistent with the cooling during the course of the Holocene. This decline in forest vegetation is representative for the entire Asian region (Fig. 5b). In contrast, the biome reconstructions reveal a minimum in mean openness between 9 ka and 6 ka. Under the assumption that forests directly follow summer temperatures, an early Holocene forest maximum should be expected, similar to the simulation. This implies a multi-millennial disequilibrium of the (Asian) forest biome to the continental climate change as previously proposed by Herzschuh et al.[49].

## Implications of the temporal mismatch for model evaluation and future vegetation change

The temporal differences between the simulated and reconstructed NH forest expansion during the deglaciation suggest that the hemispheric mean forest dynamics or at least the pollen-based reconstructions may lag the climate signal by several thousand years. The reasons for this discrepancy can be manifold. Several processes important for vegetation colonisation and migration are not implemented in the model. The strongest mismatch occurs in Asia for which the lowest amount of data is available, leaving methodological origins of the model-data differences open. So far, we can only exclude that the mismatch is caused solely by biases in the climate since the long-term simulated climate dynamics in MPI-ESM are consistent with pollen-independent reconstructions.

Our results raise the questions on which time scales pollen-based climate reconstructions are reliable. The assumption of equilibrium between continental-scale vegetation and climate in model-

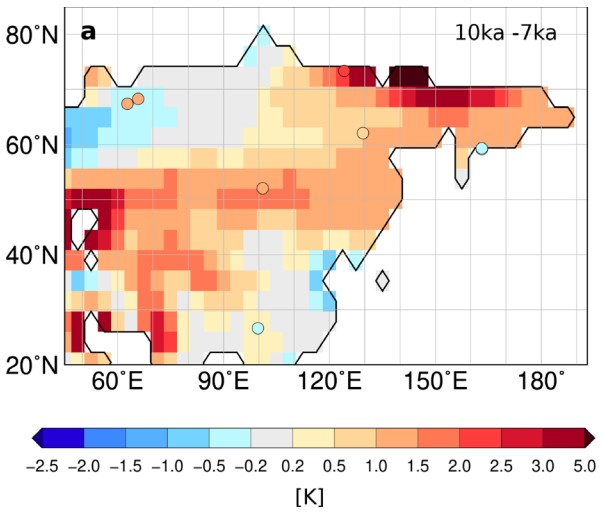
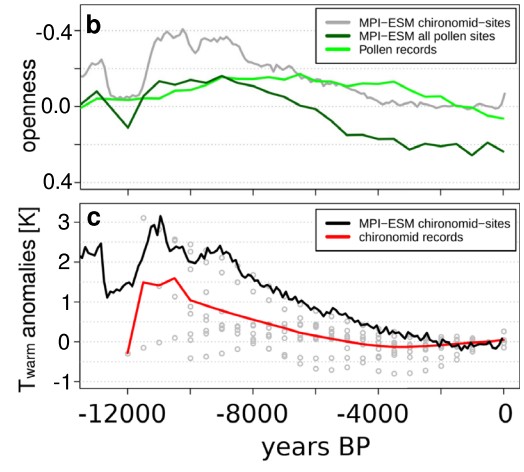

**Fig. 5 | Comparison of the simulated temperature with chironomid-based reconstructions for Asia.** The simulated July temperature is compared to chironomid-based July temperature reconstructions[60]. Only reconstructions that cover also the Early-Holocene have been used and interpolated on equal distant time-steps of 500 years. Shown are (**a**) the difference between the 10 ka and 7 ka time slices in the model (shaded) and the reconstructions (dots), prepared with the software GMT 5.4.3[80]. **b** The normalised openness index (cf. Figure 3) as simulated mean over all model grid-cells in which a chironomid-site is located (grey) or a pollen-record is located (dark green) and as mean over all Asian pollen sites; and (**c**) the simulated (black) and reconstructed mean July temperature as difference to 0 ka. In addition, all individual records are displayed (grey dots). Please note, that for 12 ka only one reconstruction is available, forming the sharp temperature increase out of the Younger Dryas in the reconstructions.

data comparisons for time-slices needs to be critically re-assessed with independent proxies. So far, it is not known to what extent the mismatch in hemispheric mean forest dynamic is affecting the pollen-based climate reconstructions. Therefore, we suggest using pollen-based climate reconstructions with great care.

Although past vegetation distributions with open landscapes and trees in refugia substantially differ from today's situation with widespread species distribution and massive human intervention, the results of this study may point to a problem in the ongoing and future anthropogenic climate change debate. Within the next 100–150 years, vegetation is projected to face a global mean temperature rise comparable to the magnitude of the warming during the deglaciation, but on much shorter timescales[7]. Though the rate of vegetation change has increased since the late Holocene[61], the change in plant distributions is expected to lag this rapid climate change[8,9], leading to strong structural and compositional transformations and consequences for the functioning of ecosystems[62]. Our study suggests that the NH forest biome in its entirety may respond much slower to current and future ongoing climatic changes than Earth System Models which lack important processes to represent these dynamics, project. This may result in false projections of the vegetation cover and misrepresentations of the terrestrial carbon storage and vegetation-atmosphere interactions.

## Methods

### MPI-ESM and the dynamic vegetation module

The transient model experiment has been performed with the Max Planck Institute Earth System Model (MPI-ESM), version 1.2[63], in spatial resolution T31GR30. The model includes the land surface model JSBACH3[64] that incorporates a dynamic vegetation module[34,65]. Vegetation is represented in the form of plant functional type (PFT) cover fraction per grid-cell. In this simulation, only natural vegetation (i.e., not affected by humans) is considered. The model distinguishes extratropical and tropical trees that can either be evergreen or deciduous, raingreen and cold-resistant shrubs, and C3 and C4 grasses. The occurrence of each PFT is constrained by temperature thresholds that reflect the bioclimatic tolerance (e.g., cold resistance, chilling or heat requirements) of the plants.

The dynamics of changes in the PFT cover fraction in response to climate change are further driven by the relative differences in annual net primary productivity (NPP) between the PFTs, which depends for instance on the moisture availability and requirements of the plants[34]. This mechanism reflects the competition between plants in a climate environment suitable for several PFTs. Generally, woody PFTs are favoured over grasses. The fraction of PFTs is furthermore reduced by disturbances such as fire and wind-throw or natural mortality. JSBACH calculates the fraction of grid-cell covered by bare ground via the carbon dynamics.

The version of JSBACH used in this study includes neither a permafrost model nor a dynamic soil buildup model nor a plant dispersal model. Soils cannot change and do not have to build up, if land is exposed due to melting ice-sheets or lifting shelf areas. Seeds for all PFTs are assumed to exist everywhere so that the vegetation dynamics solely depend on simulated climate change. Thus, the vegetation module in JSBACH, like most other models coupled to Earth System Models, determines potential vegetation in a quasi-equilibrium with climate[46]. The timescales of PFT establishment are short and mainly depend on the time scales involved in the NPP dynamics (allocation, perturbation and mortality). The time scales of mortality and allocation range from 1 year (grass) to 50 years (extratropical trees). The establishment or decay of PFTs can take place within decades[34]. Thereby, long-term simulations with DGVMs in principle provide a way to assess how strongly the simulated and reconstructed rates of vegetation change deviate and to what extent the real vegetation or pollen-based biome reconstructions are in disequilibrium with the general (simulated) climate trend. This furthermore provides constraints for the timescales on which pollen-based climate reconstructions are reliable.

### The transient simulation setup

We have conducted a transient model experiment from 22 ka to 0 ka. In this experiment, we prescribed the orbital forcing from Berger[66] and greenhouse gas forcings from Köhler et al.[67]. Orbital parameters and greenhouse gas concentrations were supplied to the model as 10-year mean values and were updated every 10 model years. Atmospheric aerosols were held constant at 1850 conditions, and we considered

natural vegetation only, thus ignoring anthropogenic land use. We prescribed the ice sheet extent from the GLAC-1D ice sheet reconstruction[68].

We conducted a model spinup of several millennia duration at constant 26 ka boundary conditions with a slightly older model version, which, due to a bug in the topography code, had a slightly warmer glacial model state than the model version used in the present experiments. We thus initialised the transient experiment at 26 ka from slightly too warm boundary conditions and ran the model transiently from then to 0 ka, i.e., the year 1950 CE. The model temperatures took about 3000 years to adjust. We therefore limited our analysis to the years after 22 ka. Ice sheet extent, as well as bathymetry and topography[69], and river routing[70] were continuously updated in ten-year intervals throughout the deglaciation.

Kapsch et al.[71] describe the general dynamics of transient climate change experiments in this setup and the effect of the choice of ice sheet forcing. In particular, they found that the meltwater forcing derived from the GLAC-1D ice sheet reconstruction leads to a collapse of the Atlantic Meridional Overturning Circulation (AMOC) from Meltwater Pulse (MWP) 1a at 14.5 ka, early in the Bølling-Allerød warm period, leading to an immediate cooling of the areas surrounding the North Atlantic.

In order to obtain a climate substantially closer to climate reconstructions, we thus modified the meltwater runoff from the Laurentide ice sheet. From 15.2 ka to 12.8 ka, we removed the meltwater flux from the Laurentide ice sheet, preventing it from affecting the AMOC and instead accumulating it, thus mimicking the meltwater storage in proglacial lakes like Lake Agassiz. Starting at 12.8 ka, we finally released the accumulated meltwater over a period of 1000 years into the Mackenzie River basin[72], which due to the changing river routing became the main outflow of the Laurentide ice sheet meltwater by that time. After reaching the Arctic ocean, this meltwater release then leads to a collapse of the AMOC and a substantial cooling of the NH areas around the North Atlantic, precisely at the time of the Younger Dryas cold reversal. Once the meltwater release is stopped at 11.8 ka, climate recovers quickly. The trend in temperature from the Bølling/Allerød to Early Holocene simulated in this way fits well to the reconstructed temperature evolution in Greenland (Fig. 1d) and Europe (Ammersee d18O, Supplementary Fig. 1a)

In this study, the simulation output has been averaged in 100-year climatological means, i.e., the first time step at 21.95 ka is an average of the years 22000- 21901 years ago. The second time step (21.85 ka) is an average of the years 21900-21801, etc. The simulation ends at the year 2000 CE.

## Evaluation of the simulated climate

Most proxy records in our simulation validation are not available in temperature or precipitation units. Therefore, we quantify the similarity between our simulation, TraCE-21 ka simulation, and the proxy records based on the similarity of the temporal patterns of the time series. We use Gaussian kernel correlation (GKC) which is a correlation-based similarity estimator for time series with irregular time axis[73]. The correlations are mostly very high (cor > 0.8) showing the good agreement of the deglacial warming trends in our simulation and the proxy records (Supplementary Table 2). As the deglacial warming trend is the dominant temporal pattern in most time series, we additionally isolate millennial-scale temporal patterns (e.g., imprints of the Bølling/Allerød and Younger Dryas) using Gaussian smoothers. The millennial-scale correlations are lower than for the orbital trends but mostly significant and of comparable magnitude to the TraCE-21ka simulation.

## Biomisation of the model results and biome reconstructions

The simulated PFT cover fractions are converted into mega-biomes using the tool by Dallmeyer et al.[36]. This method only requires a few bioclimatic limitation rules and assumptions on the minimum coverage of certain PFTs that are needed to assign the PFT composition to a biome. The grouping into mega-biomes follows the definitions commonly applied for pollen-based biome reconstructions. Tropical and warm-temperate forest biomes can only occur, if the simulated tree cover fraction exceeds the grass cover and desert fraction. For temperate and boreal forests, woody PFT cover fractions larger than 25% and a total vegetation cover of more than 50% are needed. The kind of forest is than determined by temperature thresholds. Forests are considered first, afterwards the remaining area is tested for fulfilling the constraints for the other biomes.

The method reveals similarly good results as climate-based biomisation techniques and has been tested for different time-slices and Dynamic Global Vegetation Models[36], among those the simulations with dynamic vegetation conducted in the Palaeoclimate Modelling Intercomparison Project Phase III[74] (PMIP3). With a best neighbour score (BNS) of 0.51 for 21 ka, 0.57 for 6 ka and fractional skill score (FSS) of 0.03 for PI (Supplementary Fig. 5), time-slices of the MPI-ESM simulation used here indicate similar agreement to the reference datasets as the set of simulations tested in Dallmeyer et al.[36]. Their BNS ranges from 0.25 to 0.57 for 21k and from 0.44 to 0.72 for 6 ka. The FSS for the pre-industrial time-slice (PI) reaches values between 0.0 and 0.27 (Supplementary Fig. 5). For all time-slices and all individual PFTs, this MPI-ESM simulation reproduces the reference dataset and reconstructions as well as the other simulations. The temperate forest is even better captured than in most other simulations. For the 21 ka time-slice, this simulation is among the best of the tested simulations. Overall the desert and savanna biome are the worst represented, but they do occur only at few sites north of 30°N.

One possible source of error leading to disagreement between the models and the reconstructions may be the bioclimatic thresholds in the models and the biomisation method. These thresholds are derived from modern climatic conditions and are assumed to be independent of climate. This assumption may not hold for climates that strongly differ from present-day climate.

In this study, the synthesis of biome reconstructions for the Northern Hemisphere (north of 30°N) prepared by Cao et al.[28] is used. It covers the last 40000 years and is based on fossil pollen records available in the Neotoma palaeoecology database (www.neotomadb. org). The pollen dataset has been taxonomically harmonised and temporally standardised for each continent and has been tested to ensure the quality of the data[28,75]. Pollen abundances of all available pollen assemblages have been averaged within equally distant time windows (length: 500 years back to 12 ka, 1000 years back to 21 ka, and larger windows before) and assigned to target time-slices. Standard biomisation techniques have been used to assign pollen taxa to PFTs and afterwards these PFTs to biomes. For each site and each biome, affinity scores have been calculated indicating the agreement of the PFT composition to the pre-defined biomes[35]. The biome with the highest affinity is taken as the actual biome. To allow for deviations in the properties of the pollen data and taxa to PFT and PFT to biome assignments, the biomisation is performed for each continent separately. Thereby, *Larix* pollen abundances have been multiplied by a factor of 15 to overcome low pollen productivity rates (following Bigelow et al.[76]). The record density on the different continents differ substantially. For Asia, 418 records have been used. North America has been represented by 728 sites. For Europe, 741 records have been chosen. For further details on the biomisation of the pollen records we refer to Cao et al.[28]. In this study, 81 outlier sites from this dataset have been excluded. We define sites as critical for which the simulated biome deviate from the modern biome estimate[77] and from the biome reconstructions.

## Construction of spatially averaged biome curves

The biome reconstructions face the problem of regional and temporal imbalances in the record density. The number of sites strongly

increases during the deglaciation, overlapping the forest cover trend. Most records cover only parts of the simulation period. In addition, much less sites exist for Asia than for Europe or North America, where particularly the temperate forest zone is represented by a large number of records. This may lead to a systematic misrepresentation of the area covered by forests in the reconstructions, although the Asian dataset is accepted as spatially and temporally representative of regional vegetation changes[43,75].

To partly overcome this problem in record density and to facilitate a more direct model-data comparison, we construct regional forest curves (Fig. 2) from the simulated and reconstructed biome distributions on grid-cell basis: For each reconstructed sample, we create a corresponding simulated biome sample by assigning the simulated biome from the nearest grid box to the site. Thereby, we create a dataset of simulated biomes with the same spatio-temporal distribution than the reconstructed ones. To reduce the sampling bias from the non-uniform spatial distribution of sites, we then aggregate the sites into grid boxes of size 10° x 5°. Next, we compute for each time-step a continental or hemispheric average from the grid-box mean values by weighting the forest fraction with the grid-cell area. Uncertainty estimates are computed with bootstrapping for each time step. This means that we resample all sites with a biome reconstruction at a given time step with replacement. Then, the procedure described above (i.e., grid-box aggregation, area weighted mean computation) is repeated using the resampled sites. Confidence intervals are deduced from the empirical distribution of 1000 bootstrap replicates.

For the Northern Hemisphere curve, all sites north of 30°N are included. The Asian curve contains sites north of 30°N and between 50°E and 180°E, the European curve contains sites north of 30°N and between 10°W and 50°E, and the North American curve contains sites north of 30°N and between 180°W and 47°W.

## Data availability

The biomisation dataset (reconstructions and model results) and the simulated climate data used in this study have been deposited in the MPG publication repository: http://hdl.handle.net/21.11116/0000-000A-B8D5-6 and can be downloaded free of charge. The biome affinity scores and the reconstructed dominant biomes can be obtained by contacting Xianyong Cao. The chironomid-based climate reconstructions (https://lipdverse.org/Temp12k/current_version/) and the other climate reconstructions (https://www.ncei.noaa.gov/access/paleo-search/) used for the evaluation of the model results are available from public databases. The semi-quantitative moisture reconstructions for Asia are protected and are not available due to data privacy laws. The TraCE-21ka simulation performed by F. He et al. can be downloaded here: https://www.earthsystemgrid.org/dataset/ucar.cgd.ccsm.trace.html

## Code availability

Analysis and plot scripts that have been used in this study have been deposited in the MPG publication repository: http://hdl.handle.net/21.11116/0000-000A-B8D5-6.

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

## Acknowledgements

This work contributes to the project PalMod, funded by the German Federal Ministry of Education and Research (BMBF), Research for Sustainability initiative (FONA, http://www.fona.de, last access: 3rd February 2022) and the Cluster of Excellence EXC 2037 "CLICCS - Climate, Climatic Change, and Society" (Project Number: 390683824) funded by the Deutsche Forschungsgemeinschaft (DFG, German Research Foundation) under Germany's Excellence Strategy. A.D. and T.K. were financed by PalMod (grant no. 01LP1920A and 01LP1921A). N.W. acknowledges funding by the Deutsche Forschungsgemeinschaft (DFG, project no. 395588486) and by PalMod (grant no. 01LP1926C). X.C. was financed by National Natural Science Foundation of China (grant no. 41988101) and the Sino-German Mobility Programme (grant no. M-0359). This work used resources of the Deutsches Klimarechenzentrum (DKRZ) granted by its Scientific Steering Committee (WLA) under Project ID bm1030. Open Access funding was enabled and organised by the project DEAL. We thank C. Reick (MPI-M) for his helpful comments on an earlier version of the manuscript.

## Author contributions

T.K. run the simulation, X.C. provided the reconstructions, A.D., N.W. performed the analysis. All authors were involved in the discussion of the analysis. A.D., N.W., M.C., T.K. and U.H. wrote the manuscript.

## Funding

## Competing interests

The authors declare no competing interests.
