## [Peer Review File · Nature Communications]

The deglacial forest conundrumREVIEWER COMMENTS

Reviewer #1 (Remarks to the Author):

Dallmeyer et al. manuscript submitted to Nat. Comm. presents very interesting and challenging results suggesting a multi-millennial temporal mismatch between the simulated Northern Hemisphere (NH) extra-tropic forests and the pollen-based biome reconstruction for the last deglaciation. This mismatch implies that NH forests may be responding much slower to ongoing climate change than Earth System Models currently predict. This work deserves publication after the authors clarify/discuss the issues that I have listed below.

In the Introduction, the authors discuss the difficulty in assessing to what extent reconstructed vegetation is in equilibrium with climate. I think that Dallmeyer et al. should specify more clearly that they refer to global or hemispheric vegetation. Several works have shown that there is a dynamic equilibrium between regional or sub-continental scale vegetation and the rapid changing climate of the last glacial period. For example, millennial to centennial-scale changes in eastern North Atlantic sea surface temperatures are almost synchronous with vegetation changes in western Europe (Sanchez Goñi et al., 2008).

Three causes are listed by Dallmeyer et al. for explaining the temporal difference in simulated NH forest expansion related to delayed vegetation response to climate signal: a) the biomisatiion method applied to pollen assemblages with, for instance, some trees being underrepresented, b) the rapid response of the PFTs assumed by the Dynamic Global Vegetation Models, models that are moreover not implemented by soil and permafrost evolution, and c) the climate biases affecting the simulated climate change that produces stronger simulated precipitation in the East Asian core monsoon zone compared to the reconstructed one (Extended data Figure 1h).

I am wondering whether this temporal mismatch (Figure 1f), mainly caused by the expansion of the Asian forest in the model, is also due to : a) the low temporal resolution of the pollen records from Asia that would be unable to detect the millennial-scale Bølling-Allerød, Younger Dryas and onset of the Holocene at ~11 ka, and b) the stronger simulated temperature increase compared to that reconstructed from chironomids in Asia. Besides that, the explanation given by the authors to justify that vegetation delayed the climatic signal is based on the pollen-independent moisture reconstructions from speleothems. However, these reconstructions are hampered by, as clearly stated by Dallmeyer et al., the ability of these records to document rainfall.

Lines 271-277 : following the work by Loarie et al (2009) not all biomes would strongly delay the climate change. In particular, mountainous ecosystems and the temperate conifer forest belt can pace the relatively low climate change predicted in these regions. The authors should mention that. Finally, if the disequilibrium of the NH forest biome to the climate signal is real the authors can briefly discuss this issue in the light of a recent paper by Rietkerk et al. (2021). These authors propose that spatial self-organization can cause ecosystems to evade tipping points and can thereby be a signal of resilience.

Minor corrections

- Line 52: Add a dot after « alone ».
- Line 111 : Please replace « 1g » with « 1h ».
- Line 122 : Please add « at hemispheric scale » after « ...the natural vegetation... »
- Line 266 : I think that it is more appropriate to specify that the assumption of equilibrium between « global/hemispheric » vegetation and climate in model-data comparisons...
- Line 585 : Please add the meaning of the FSS abbreviation.
- Extended data Figure 1, Line 800 in the SI : please add MPI-ESM 1.2 before « The simulated climate change ».
- Extended data Figure 1, Line 805 : please indicate the meaning of the blue and pink bands.
- Extended data Figure 2 : why are not marine pollen records included ?
- Figure 3 : what is the meaning of the diamonds ?
- Extended data Figure 5, Line 860 : « c » should be in bold.
- Extended data Figure 6. Please add (a) and (b) for the left and right panels, respectively.

References

Rietkerk et al. (2021) Evasion of tipping in complex systems through spatial pattern formation. *Science* 374: 169-178.

Sánchez Goñi et al., (2008) Contrasting impacts of Dansgaard-Oeschger events over a western European latitudinal transect modulated by orbital parameters, *Quaternary Science Reviews* 27: 1136-1151.

Reviewer #2 (Remarks to the Author):

The study compares pollen and simulation-based biome and forest reconstructions for the northern Hemisphere extra-tropics considers discrepancies between the two approaches. The study is stimulating and of interest for understanding how data gaps affect large-scale analyses, particularly with the popularity of meta-analyses that combine datasets across wide areas. My review is written from the perspective of a palaeoecologist/palynologist with a strong interest in the strengths and limitations of the method, not as a palaeoclimatologist or modeller, which I do not have expertise in. I therefore focus my comments primarily on palynological aspects, not the details of the simulation model. I strongly support efforts to stress-test the ability of proxies and stimulate debate, which this study contributes to.

However, the paper would benefit from a more balanced approach to the strengths/limits of both modelling and pollen. This is especially pertinent in the abstract, which is provocative but privileges the model results over the pollen reconstruction without sufficiently representing the uncertainties and questions that apply to both approaches.

To me, the main contributions from the study are (1) that it identifies data-deficiencies in the Asian region, and (2) highlights how such gaps in data availability limit and potentially bias our understanding and reconstructions/models of how climate shapes vegetation dynamics in these areas. This contrasts markedly with the broad scale perspective suggested by the current title and abstract. I recommend revisions to the language and main messages throughout to shift the focus to these two aspects, as the current emphasis on pollen vs simulation evidence does not really stand up to scrutiny.

See attached file for specific comments.

Nature Communications review. The deglacial forest conundrum

General comments

The study compares pollen and simulation-based biome and forest reconstructions for the northern Hemisphere extra-tropics considers discrepancies between the two approaches. The study is stimulating and of interest for understanding how data gaps affect large-scale analyses, particularly with the popularity of meta-analyses which combine datasets across wide areas. My review is written from the perspective of a palaeoecologist/palynologist with a strong interest in the strengths and limitations of the method, not as a palaeoclimatologist or modeller, which I do not have expertise in. I therefore focus my comments primarily on palynological aspects, not the details of the simulation model. I strongly support efforts to stress-test the ability of proxies and stimulate debate, which this study contributes to. However, the paper would benefit from a more balanced approach to the strengths/limits of both modelling and pollen. This is especially pertinent in the abstract, which is provocative but privileges the model results over the pollen reconstruction without sufficiently representing the uncertainties and questions that apply to both approaches.

To me, the main contributions from the study are (1) that it identifies data-deficiencies in the Asian region, and (2) highlights how such gaps in data availability limit and potentially bias our understanding and reconstructions/models of how climate shapes vegetation dynamics in these areas. This contrasts markedly with the broad scale perspective suggested by the current title and abstract. I recommend revisions to the language and main messages throughout to shift the focus to these two aspects, as the current emphasis on pollen vs simulation evidence does not really stand up to scrutiny.

Specific comments

Please see annotations on ms pdf for minor comments (clarity of expression).

L70-71. "pollen data are limited in their ability to record vegetation dynamics". Yes, there certainly are interpretational caveats, but also much effort to understand and address these (e.g. Chevalier et al. 2020 ESR, Birks et al. 2010 OEJ), including many existing studies which compare process-based simulations and pollen-based vegetation reconstructions. This needs to be acknowledged. Please edit the phrasing: rigorous science should not be based on setting up a strawman argument.

Reference to geographical regions: please check and edit use of "Europe" and "Asia" for clarity (using distinctions presented in Methodology L643). I suggest using Eurasia where the wider region is meant, as this is often confusing, e.g. L135: 'Europe' is used, but I think Eurasia or even Asia is meant and would be clearer, as the main model-reconstruction difference relates to Asia.

L153 refers to "local"; in pollen studies, this generally refers to a scale of metres to a few km, which the text clearly does not mean. Please state what scale is meant.

No analogue assemblages: L168-9. Are human impacts (to the extent of decoupling vegetation-climate) the most significant factor for this area/time period? If anthropogenic factors are considered significant, please comment on the implications for using climate-based models to simulate vegetation. The model is limited to 'natural vegetation' so presumably cannot be used to comment on human impacts. There is extensive literature on no analogue climates, which could affect both the reconstructions and simulations (similar to the caveat in L593-7). E.g. Williams & Jackson 2004, 2007 FEE, Williams et al. 2013 *Annals of the New York Academy of Sciences*. Please revise to state what the key implications are for reconstruction and modelling, beyond just biome assignment, to show awareness of wider literature on the topic and to stress the need for clarity on

assumptions underpinning the methods. Some of this is in the methodology, but many of the caveats considered there, particularly relating to model parameterisation, aren't mentioned in the main text, which gives a bias towards the model findings and limits the sense of critical evaluation.

L173. How much does pollen productivity influence biome reconstructions or model responses? Low productivity for *Larix* is known (cf. Jackson 2012 QSR), so it can be taken into account when reviewing and interpreting results. E.g. Is sensitivity-testing available or required?

L191/205. In addition to the factors listed, soil development, no-analogue conditions and interactions between multiple factors could also be considered as these are not just specific to pollen/vegetation, but also have implications for the completeness of model assumptions, since soil development is explicitly excluded (L499). See Birks & Birks 2008 *Holocene* for these: especially relevant around L205 since their study provides a pollen/macrofossil comparison of successional lags that may be relevant to the Siberian example. This highlights why the model assumption that all seed sources for all PFTs exist everywhere (L501) may be inappropriate and could generate more rapid predictions than observed via pollen. Did you consider examining macrofossil records to sense-check the pollen, particularly regarding lagged appearance of species? Please revise to offer a more representative list of key factors affecting migration proposed in the wider literature.

L215. Lags of 400 years are also proposed for pioneer species: Birks & Birks 2008. Giesecke et al. (2007, 2017 *J Biogeog*) also identify continental and regional differences in migration rates, which raises questions about appropriate scales of reconstruction/modelling that are relevant to the point about transferability.

L237. Not implausible but these would be small populations which would make small pollen and seed contributions to biome reconstructions and migration. Migration rates may also have been influenced by wetness and landscape heterogeneity. Reflecting on how issues raised might influence uncertainty in both pollen and climate reconstruction/models would give a better sense of balance and integration, rather than the current organisation, which considers each method separately.

L250-4 offer a simplistic interpretation that is at odds with the range of uncertainties and potential complexities referred to in the preceding text. Please revise to offer a more nuanced interpretation, rather than privileging the model results.

L261-2. You state that the model is consistent with pollen-independent reconstructions. However, the preceding text on this is brief and the language is highly qualitative (e.g. similar, weaker), rather than offering any quantitative comparison. To establish the model as robust and rigorous, it would be useful if the comparisons could be more quantitative (e.g. refer to BNS or fss) and take into account variations in the temporal/spatial coverage of evidence (e.g. sparse evidence for Siberia: L623, and assumptions involved in spatial generalisation/averaging L635). You refer to data availability, but do not mention the data quality; both are needed to demonstrate that the reconstruction and model are robust.

L267-8. Much literature considers the issue of (dis)equilibrium but none is cited here. Please show your awareness of the existing literature and debates, beyond that of Svenning and colleagues, particularly relating to migrational lag. E.g. Williams et al. 2021 *Nature Eco Evo*.

L271. The link from the lateglacial and Holocene to future is rather abrupt. The vegetation/species distribution starting point for the future is markedly different from the lateglacial (e.g. widespread distribution of many species now, rather than lateglacial refugial starting point) and, as above, there is an extensive literature on the ability of trees to track climate change (past and future) which is not

referenced. Please review the conclusions to ensure they are better supported by the temporal focus and findings of the study.

L521. How do you define 'natural vegetation'?

Extended data figure 2. Please add what size time steps are used in (a).

Extended data figure 4. There are numerous low similarity reconstruction/simulations in the east central area of North America: is this in keeping with previous studies (which are numerous)? This better-studied area seems an ideal location to consider some of the potential uncertainties and/or unknowns, for comparison with the more data-deficient parts of Asia that form the focus of the current text. Giving a comparison would allow the paper to present a stronger argument.

Reviewer #3 (Remarks to the Author):

This manuscript presents a data-model comparison of vegetation through the LGM to present, with a focus on mis-matches between simulated vegetation via a dynamic vegetation model (JSBACH), and reconstructed vegetation, classified from pollen records (biomisation method), across the northern hemisphere (north of 30 degrees). The noteworthy result is that the evolution of JSBACH-modelled tree cover does not match biomised pollen through the LGM. The authors explain that there are really two plausible reasons why this is the case: either biomisation isn't very good, or vegetation and soil dynamics are poorly captured in ESMs.

I am not surprised by these results, and I would expect that the majority of scientists who are familiar with ecological processes would expect this result. The implication is that pollen data are not ideal for estimating climate i.e. the results of studies such as Mauri et al 2015 should be viewed with extreme caution as vegetation is not in equilibrium with climate. I feel that we have known this already for at least a decade, but perhaps this point has not been sufficiently well made in the literature. The implication is not drawn out in this study, though it is an important point to make as these data are still perhaps uncritically used.

The manuscript offers a series of problems and plausible explanations of the discrepancy, but does not offer solutions to overcome these problems. Taking first the suggestion that biomisation may not represent vegetation. A possible solution is to apply alternative methods to the pollen data. The state-of-the-art now appears to be quantitative reconstruction via methods such as REVEALS, and full Holocene reconstructions are now available for different parts of Eurasia at least (e.g. Li et al 2020; Githumbi et al 2022). These have the advantage of estimating forest fraction in grid cells, rather than the classified biomisation approach. There remain problems of parameterising these models if pollen production varies during periods of major climate upheaval, but this is a problem inherent in any analysis of pollen data. The authors might reflect on this as a direction of travel to overcome the problem that is faced (or at least a way of testing whether it is the biomisation process itself that causes the mis-match).

I agree entirely with the authors that vegetation dynamics are not well captured in the ESM. But the authors do not suggest a solution, only point out the problem. In my view what is needed is a thorough review of possible rates of migration of different tree species (note: not all trees are the same), placed within a "velocity of climate change" framework. The climate simulations should enable us to calculate velocities of climate change (which vary across time and space, and are strongly controlled by eg. topography), and comparison can be made with the possible velocities of plant migration (for different tree species). This would enable us to assess whether trees kept up with, or significantly lagged, climate in the past, as a lesson for the present and future.

This point is important in the implications of the study, that trees may not track future climate. Line 271 implies that current climatic shifts will be stronger than during deglaciation. I strongly question this statement: the authors declare a 5-7K shift across the LGM, which is understood to have taken place in < 50 years (Steffenson et al 2008). Current disruptions to the climate system are not on that order of magnitude. I agree that future change is alarming and of major societal concern, but the statement needs refining.

A wider question that should be addressed is the implication for land cover feedbacks in ESMs. If DVMs are a component of the ESMs and feedback to force the climate system, then if these are not correct then that has major implications for the validity of results of ESMs. Work is now showing that land cover plays an important forcing role in the climate system (Strandberg et al 2022). The authors may want to reflect this in their discussions.

The methodologies that are used in the paper are sound, and descriptions are adequate to reproduce the work.

Mauri, A., Davis, B. A. S., Collins, P. M., & Kaplan, J. O. (2015). The climate of Europe during the Holocene: a gridded pollen-based reconstruction and its multi-proxy evaluation. *Quaternary Science Reviews*, 112, 109-127.

Strandberg, G., Lindström, J., Poska, A., Zhang, Q., Fyfe, R., Githumbi, E., ... & Gaillard, M. J. (2022). Mid-Holocene European climate revisited: New high-resolution regional climate model simulations using pollen-based land-cover. *Quaternary Science Reviews*, 281, 107431.

Li, F., Gaillard, M.-J., Cao, X., Herzschuh, U., Sugita, S., ..(18).. and Jia, X. (2020). Towards quantification of Holocene anthropogenic land-cover change in temperate China: A review in the light of pollen-based REVEALS reconstructions of regional plant cover. *Earth-Science Reviews*, 203:103119

Steffensen, J. P., Andersen, K. K., Bigler, M., Clausen, H. B., Dahl-Jensen, D., Fischer, H., ... & White, J. W. (2008). High-resolution Greenland ice core data show abrupt climate change happens in few years. *science*, 321(5889), 680-684.

Strandberg, G., Lindström, J., Poska, A., Zhang, Q., Fyfe, R., Githumbi, E., ... & Gaillard, M. J. (2022). Mid-Holocene European climate revisited: New high-resolution regional climate model simulations using pollen-based land-cover. *Quaternary Science Reviews*, 281, 107431.

Reply to REVIEWER COMMENTS: “The deglacial forest conundrum”

Reviewer #1 (Remarks to the Author):

Dallmeyer et al. manuscript submitted to Nat. Comm. presents very interesting and challenging results suggesting a multi-millennial temporal mismatch between the simulated Northern Hemisphere (NH) extra-tropic forests and the pollen-based biome reconstruction for the last deglaciation. This mismatch implies that NH forests may be responding much slower to ongoing climate change than Earth System Models currently predict. This work deserves publication after the authors clarify/discuss the issues that I have listed below.

Reply: We would like to thank the reviewer for his/her effort and encouragement of our work. We appreciate very much all suggestions and we incorporate most of them in the revised manuscript. The reviewer comments are shown in black, our answers in blue.

In the Introduction, the authors discuss the difficulty in assessing to what extent reconstructed vegetation is in equilibrium with climate. I think that Dallmeyer et al. should specify more clearly that they refer to global or hemispheric vegetation. Several works have shown that there is a dynamic equilibrium between regional or sub-continental scale vegetation and the rapid changing climate of the last glacial period. For example, millennial to centennial-scale changes in eastern North Atlantic sea surface temperatures are almost synchronous with vegetation changes in western Europe (Sanchez Goñi et al., 2008).

Reply: We thank the reviewer very much for this suggestion. Following this comment, we emphasize more in the manuscript that it is the hemispheric wide, or rather the continental-wide vegetation that is probably not in equilibrium with climate.

Three causes are listed by Dallmeyer et al. for explaining the temporal difference in simulated NH forest expansion related to delayed vegetation response to climate signal: a) the biomisatiion method applied to pollen assemblages with, for instance, some trees being underrepresented, b) the rapid response of the PFTs assumed by the Dynamic Global Vegetation Models, models that are moreover not implemented by soil and permafrost evolution, and c) the climate biases affecting the simulated climate change that produces stronger simulated precipitation in the East Asian core monsson zone compared to the reconstructed one (Extended data Figure 1h).

I am wondering whether this temporal mismatch (Figure 1f), mainly caused by the expansion of the Asian forest in the model, is also due to : a) the low temporal resolution of the pollen records from Asia that would be unable to detect the millennial-scale Bølling-Allerød, Younger Dryas and onset of the Holocene at ~11 ka, and b) the stronger simulated temperature increase compared to that reconstructed from chironomids in Asia.

Reply: The temporal resolution of the Asian records is the same as for the other continents (~500 years during the deglaciation and the Holocene). In this resolution and averaged over all sites (Fig 1f), the Bølling-Allerød / Younger Dryas

transition can not be represented, but the beginning of the Holocene should be represented, at least more or less with +/-500 years. We do not think, that this can explain a delay by approximately 4000 years. We furthermore see (Fig.2) that the maximum expansion speed is fairly similar between reconstructions and simulation. It just starts much later in the records (and also ends later). This is not a behavior you would expect if low resolution of records would drive the mismatch. Low resolution would just smooth the curve (i.e. the expansion would start earlier and end later in the records, which is not the case).

The stronger temperature difference between early-Holocene and 0ka in the model compared to the chironomid-based reconstructions may imply a larger tree cover in the model compared to “reality”. However, we do not expect an effect on the temporal change. To further strengthen our conclusion, we added the simulated and reconstructed vegetation to Figure 3 in form of the openness-index already used in the manuscript. The simulated tree cover at the chironomid-sites is indeed higher compared to the mean over Asia, but the temporal change is in line, showing a minimum in openness (i.e. maximum in trees) around 11ka-10ka. In contrast, the reconstructions show the minimum in openness around 6.5ka.

We furthermore add to the text (L337-330): *“The simulated decrease in openness at sites with chironomid-based reconstructions is in line with the temperature decline and is representative for the entire Asian region (Figure 5b), whereas the reconstructions reveal a minimum in mean openness between 9ka and 6ka.”*

“Caption Fig.3: **Figure 5**

Comparison of the simulated temperature with chironomid-based reconstructions for Asia

The simulated July temperature is compared to chironomid-based July temperature reconstructions⁶⁰. Only reconstructions that cover also the Early-Holocene have been used and interpolated on equal distant time-steps of 500 years. Shown are **a)** the difference between the 10 ka and 7 ka time slices in the model (shaded) and the reconstructions (dots); **b)** the normalised openness index (cf. Extended Figure 3) as simulated mean over all grid-cells in which a chironomid-site is located (grey) or a pollen-record is located (darkgreen) and as mean over all Asian pollen sites; and **c)** the simulated (black) and reconstructed

mean July temperature as difference to 0 ka. In addition, all individual records are displayed (grey dots). Please note, that for 12 ka only one reconstruction is available, forming the sharp temperature increase out of the Younger Dryas in the reconstructions. “

Besides that, the explanation given by the authors to justify that vegetation delayed the climatic signal is based on the pollen-independent moisture reconstructions from speleothems. However, these reconstructions are hampered by, as clearly stated by Dallmayer et al., the ability of these records to document rainfall.

Reply: This is a good point. Most reconstructions shown in the Extended Figure 1 are indeed based on speleothems, because these are still one of the main alternative terrestrial archives. We know about the discussion on the uncertainties with respect to what is actually recorded by speleothems. Since we are no expert in that field, we rely on others and used these reconstructions such as stated in the literature. For the comparison of pollen-based and non-pollen based moisture reconstructions, not only speleothems have been used (other proxies: e.g. carbon to nitrogen contents, chironomids, magnetic susceptibility,...)

Lines 271-277 : following the work by Loarie et al (2009) not all biomes would strongly delay the climate change. In particular, mountainous ecosystems and the temperate conifer forest belt can pace the relatively low climate change predicted in these regions. The authors should mention that.

Reply: Thank you, this was indeed not differentiated well enough. We further specify our conclusions (L360-373): *“Within the next 100-150 years, vegetation is projected to face a global mean temperature rise comparable to the magnitude of the warming during the deglaciation, but on much shorter time-scales⁷. Though the rate of vegetation change is increasing since the late Holocene⁶¹, the change in plant distributions is expected to lag this rapid climate change^{8,9}, leading to strong structural and compositional transformations and consequences for the functioning of ecosystems⁶². Our study suggests that the NH forest biome in its entirety may respond much slower to current and future ongoing climatic changes than Earth System Models which lack important processes to represent these dynamics, project. This may result in false projections of the vegetation cover and misrepresentations of the terrestrial carbon storage and vegetation-atmosphere interactions.”*

Finally, if the disequilibrium of the NH forest biome to the climate signal is real the authors can briefly discuss this issue in the light of a recent paper by Rietkerk et al. (2021). These authors propose that spatial self-organization can cause ecosystems to evade tipping points and can thereby be a signal of resilience.

Reply: This is indeed an interesting aspect, but since in our simulation we do not see a tipping point in the spread of extratropical forests (the processes are too slow), we do not consider how we could prevent tipping of the forests. Therefore, we will not include it in this manuscript.

Minor corrections

- Line 52: Add a dot after « alone » A: done

- Line 111 : Please replace « 1g » with « 1h ». A: “1g” (the record Lianhua) is

correct, but as it is rather at the northern boundary of the EASM, we now write: ..."the northern part of the East Asian summer monsoon".

- Line 122 : Please add « at hemispheric scale » after « ...the natural vegetation... » A: done.

- Line 266 : I think that it is more appropriate to specify that the assumption of equilibrium between « global/hemispheric » vegetation and climate in model-data comparisons... A: done.

- Line 585 : Please add the meaning of the FSS abbreviation. A: We have added the abbreviation in line 496, where we use the term "fractional skill score" for the first time.

- Extended data Figure 1, Line 800 in the SI : please add MPI-ESM 1.2 before « The simulated climate change ». A: done.

- Extended data Figure 1, Line 805 : please indicate the meaning of the blue and pink bands. A: done. We have copied the text from Fig.1.

- Extended data Figure 2 : why are not marine pollen records included ? A: The Cao et al dataset only includes the records on land (north of 30°N). In addition, biomisation methods have mostly been developed and tested for terrestrial records. As archival processes are different between marine and terrestrial records, re-calibration of the methods may be necessary for marine records. However, tuning specifically for marine records is difficult due to the small number of available modern samples from marine sites. (e.g. Adam, Weitzel and Rehfeld 2021, <https://doi.org/10.1029/2021PA004265>).

- Figure 3 : what is the meaning of the diamonds ? A: You probably mean the two grid-cells looking like diamonds because the land-sea mask contour is drawn interpolated around the grid cells. These grid-cells are Japan and part of the Caspian Sea.

- Extended data Figure 5, Line 860 : « c » should be in bold. A: done.

- Extended data Figure 6. Please add (a) and (b) for the left and right panels, respectively. A: done.

References

Rietkerk et al. (2021) Evasion of tipping in complex systems through spatial pattern formation. *Science* 374: 169-178.

Sánchez Goñi et al., (2008) Contrasting impacts of Dansgaard-Oeschger events over a western European latitudinal transect modulated by orbital parameters, *Quaternary Science Reviews* 27: 1136-1151.

Reviewer #2 (Remarks to the Author):

The study compares pollen and simulation-based biome and forest reconstructions for the northern Hemisphere extra-tropics considers discrepancies between the two approaches. The study is stimulating and of interest for understanding how data gaps affect large-scale analyses, particularly with the popularity of meta-analyses that combine datasets across wide areas. My review is written from the perspective of a palaeoecologist/palynologist with a strong interest in the strengths and limitations of the method, not as a palaeoclimatologist or modeller, which I do not have expertise in. I therefore focus my comments primarily on palynological aspects, not the details of the simulation model. I strongly support efforts to stress-test the ability of proxies and stimulate debate, which this study contributes to. However, the paper would benefit from a more balanced approach to the strengths/limits of both modelling and pollen. This is especially pertinent in the abstract, which is provocative but privileges the model results over the pollen reconstruction without sufficiently representing the uncertainties and questions that apply to both approaches.

Reply: We thank the reviewer for the constructive comments. We regret that we might have provoked the impression to privilege model results over pollen reconstruction. We reformulate the Abstract to provide a more balanced view. Now the Abstract reads:

“Forest expansion in the Northern Hemisphere extra-tropics during the deglaciation occurs approximately 4000 years earlier in a transient MPI-ESM1.2 simulation than shown by pollen-based biome reconstructions. This discrepancy challenges the paleo-climate community with a new conundrum. Shortcomings in the model and the reconstructions could both contribute to this mismatch, leaving the underlying causes unresolved. The vegetation simulated in Earth System Models (ESMs) responds to climate change within decades. As the simulated climate is consistent with pollen-independent reconstructions, we can exclude climate biases as main driver for the temporal differences in forest cover. Instead, the model-data mismatch points at a multi-millennial disequilibrium of the Northern Hemisphere forest biome to the climate signal. Therefore, the evaluation of time-slice simulations in a strongly changing climate with pollen records should be critically reassessed. Our results imply that Northern Hemisphere forests may be responding much slower to ongoing climate changes than ESMs predict.”

To me, the main contributions from the study are (1) that it identifies data-deficiencies in the Asian region, and (2) highlights how such gaps in data availability limit and potentially bias our understanding and reconstructions/models of how climate shapes vegetation dynamics in these areas. This contrasts markedly with the broad scale perspective suggested by the current title and abstract. I recommend revisions to the language and main messages throughout to shift the focus to these two aspects, as the current emphasis on pollen vs simulation evidence does not really stand up to scrutiny.

This study here is meant as an introduction to a new problem in model-data comparison and we just want to draw the attention on this conundrum to encourage other groups to research in this respect as well. There are lots of

possible causes on the modeling and reconstruction side that may lead or contribute to the temporal mismatch in forest dynamics. Most of these possible causes can not be tested at the moment as we do not have the model-setups to do it, and it will take some time to prepare the models to e.g. implement dynamic soils. We would like to keep the broader overview on the existing problem as we cannot prove that the low record density in Asia causes the model-data mismatch or whether other factors drive the mismatch.

The low record density for the Asian region may indeed be problematic, as individual records are given a strong weighting in the averaging over the grid-cells. On the other hand, the forest biomes are more homogeneous in Siberia and thus could be represented by a lower number of records. The records are relatively consistent. We have tested different aggregation methods to calculate the mean forest biome change (e.g. based on sites or smaller grid-cell sizes), but the mismatch is robust.

To account for data-gaps as the possible cause of the model-data mismatch, we include this problem in the revised manuscript by writing:

L193-196: “The conundrum could be caused by (a) shortcomings in biomisation techniques and/or insufficient raw data (e.g. the low record density in Asia), (b) oversimplified vegetation and soil dynamics in ESMs or (c) biases in the simulated climate.”

*L222-231: “Particularly for Asia, the amount of high-quality pollen records are low. Hardly no sites exist in the central parts of Eastern Siberia and the records are short in the Russian Far East (**Supplementary Fig. 2**). The low data coverage may also imply that the biomisation methods are less calibrated in this region. Although the aggregation of pollen sites into grid-cells partly overcomes the problem of imbalanced record densities, the poor data coverage in Asia may affect the calculated mean forest cover change. Since the Siberian vegetation is more homogenous than on other continents and mega-biome belts are large in extent, the few records can nevertheless be representative at regional-to-continental scale⁴³.”*

L343-346: “Several processes important for vegetation colonisation and migration are not implemented in the model. The strongest mismatch occurs in Asia for which the lowest amount of data is available, leaving methodological origins of the model-data differences open.”

See attached file for specific comments.

Specific comments

Please see annotations on ms pdf for minor comments (clarity of expression).

L70-71. “pollen data are limited in their ability to record vegetation dynamics”. Yes, there certainly are interpretational caveats, but also much effort to understand and address these (e.g. Chevalier et al. 2020 ESR, Birks et al. 2010 OEJ), including many existing studies which compare process-based simulations and pollen-based vegetation reconstructions. This needs to be acknowledged. Please edit the phrasing: rigorous science should not be based on setting up a strawman argument.

Reply: We apologise for giving the impression that we see the causes for the mismatch entirely in the pollen data or biome reconstructions. That is definitely not what we wanted to say. We change this sentence to:

L97-103: "Complicating this, pollen data are in some aspects limited in their ability to record vegetation dynamics²⁶, although much effort is done to overcome these caveats². For instance, pollen productivity can be strongly decreased under lower CO₂-levels and temperatures²⁷ and the coupling of vegetation to environmental factors is complicated, challenging the reliability of pollen-based vegetation reconstructions for glacial climates."

Reference to geographical regions: please check and edit use of "Europe" and "Asia" for clarity (using distinctions presented in Methodology L643). I suggest using Eurasia where the wider region is meant, as this is often confusing, e.g. L135: 'Europe' is used, but I think Eurasia or even Asia is meant and would be clearer, as the main model-reconstruction difference relates to Asia.

Reply: We apologize if our nomenclature is misleading. We adapt the geographical extent of the regions from the pollen-dataset. The division of Europe and Asia at 50°N has already been used by Cao et al. 2019. As for each continent another biomesation procedure has been used, we would like to keep this nomenclature to be consistent to the reconstructions. We check the manuscript carefully to ensure usage of the continent-names to be consistent with this definition throughout the manuscript.

L153 refers to "local"; in pollen studies, this generally refers to a scale of metres to a few km, which the text clearly does not mean. Please state what scale is meant.

Reply: Thank you, this was indeed misleading. We change the sentence to (L183-186): *"Despite regional model-data mismatches in North America and Europe, the simulated large-scale dynamics on both continents are mostly consistent with the reconstructions."*

No analogue assemblages: L168-9. Are human impacts (to the extent of decoupling vegetation- climate) the most significant factor for this area/time period? If anthropogenic factors are considered significant, please comment on the implications for using climate-based models to simulate vegetation. The model is limited to 'natural vegetation' so presumably cannot be used to comment on human impacts. There is extensive literature on no analogue climates, which could affect both the reconstructions and simulations (similar to the caveat in L593-7). E.g. Williams & Jackson 2004, 2007 FEE, Williams et al. 2013 Annals of the New York Academy of Sciences. Please revise to state what the key implications are for reconstruction and modelling, beyond just biome assignment, to show awareness of wider literature on the topic and to stress the need for clarity on assumptions underpinning the methods. Some of this is in the methodology, but many of the caveats considered there, particularly relating to model parameterisation, aren't mentioned in the main text, which gives a bias towards the model findings and limits the sense of critical evaluation.

Reply: We do not assume that the model-data mismatch originates from human impact on forests. The method of biomisation is rather robust against human impact, because it does not directly relate modern pollen assemblages from human-biased vegetation to fossil pollen records. It assigns the taxa to PFTs and then the PFTs to biomes.

It was not our intention to prioritize the model results. To give a more balanced view, we also discuss problems with the biomisation technique in this paragraph (L203-210):

“Similarly, the model-based biomisation technique relies on only few PFTs and constant cover fraction limits and bioclimatic thresholds that have been defined from modern observations and may thus not be valid for strongly deviating climates. The PFT-biomisation procedure is rather simple. It competes with other model biomisation techniques, but the e.g. simulated biome distribution slightly deviates from modern potential biome estimates, revealing caveats of the PFT-biomisation technique³⁶ (Supplementary Fig. 5)”

Following the Reviewers suggestion we broaden the discussion on problems with the biomisation technique (L197-203):

“The pollen-based biomisation method is validated by comparisons between reconstructed biomes and modern vegetation distributions^{35,36}. However, fossil pollen assemblages may lack modern analogues for example in heavily human-altered landscapes³⁷ or because past climate and atmospheric compositions differ from modern conditions³⁸. This may lead to less reliable taxa-to-PFT and PFT-to-biome assignments, that furthermore depend at least partly on subjective choices³⁹.”

L173. How much does pollen productivity influence biome reconstructions or model responses? Low productivity for Larix is known (cf. Jackson 2012 QSR), so it can be taken into account when reviewing and interpreting results. E.g. Is sensitivity-testing available or required?

Reply: The low pollen productivity of Larix is already considered in this biomisation technique. Larix pollen abundances have been multiplied by a factor of 15 following previous studies to overcome low pollen productivity rates. For other taxa, no weighting has been applied. We agree that we have not been precise enough on this point. Pinus is a strong pollen producer and expands relatively late. We include this in the revised manuscript by stating (L219-221):

“In contrast, pine is a strong pollen producer and expands relatively late in Asia⁴¹ and may thus bias the mean forest cover change.”

L191/205. In addition to the factors listed, soil development, no-analogue conditions and interactions between multiple factors could also be considered as these are not just specific to pollen/vegetation, but also have implications for the completeness of model assumptions, since soil development is explicitly excluded (L499). See Birks & Birks 2008 Holocene for these: especially relevant around L205 since their study provides a pollen/macrofossil comparison of successional lags that may be relevant to the Siberian example. This highlights why the model assumption that all seed sources for all PFTs exist everywhere (L501) may be inappropriate and could generate more rapid predictions than observed via pollen. Did you consider examining macrofossil records to sense-

check the pollen, particularly regarding lagged appearance of species? Please revise to offer a more representative list of key factors affecting migration proposed in the wider literature.

Reply: Since the Earth System Model MPI-ESM only considers 8 different plant functional types, we do not expect that the “no-analogue” problem directly affects the model results. We add a remark in the revised main text that the bioclimatic limits in the model are based on modern observations and that this may lead to problems (L.203-210). The lack of soil development in the model has already been discussed in the text (L270-278). We agree that the “seeds are available everywhere” approach in Earth System Models imply a faster response to climate changes than reconstructed. This assumption is of course inappropriate – at least in some regions – but it is (still) standard in the vegetation models that are typically coupled in Earth System Models. First efforts are made to consider seed dispersal in (global) vegetation models (e.g. Snell, 2013, <https://doi.org/10.1111/geb.12106>), but so far MPI-ESM has no seed dispersal routine (and neither of the other models participating in PMIP4 that run the last deglaciation has one).

We have checked the occurrence of *Larix* in Siberia with the macrofossil synthesis by Binney et al. (2009, <https://doi.org/10.1016/j.quascirev.2009.04.016>). *Larix* occurred relatively early, but there are only few sites available. That’s why we decided to not include it in this manuscript.

We revise this section and add the following processes affecting migration:

L245-247: “The spatial resolution of the simulation used here is coarse, leading to an underestimation of the orography which may, in reality, act as barrier for seed dispersal.”

L255-258: “For Siberia, a misadaptation of the trees in the refugia is discussed, which would have necessitated a reinvasion of the trees from southern populations⁵⁰. This could explain the slow response in the reconstructions compared to the fast response in the model lacking a dispersal routine.”

L259-262: “Among other factors (for an overview see Williams et al.¹⁸), the expansion of plant species depends on their population growth rate and population size, local evolutionary adaption⁵², interspecific competition⁵³ or high dependencies on other species.”

L215. Lags of 400 years are also proposed for pioneer species: Birks & Birks 2008. Giesecke et al. (2007, 2017 J Biogeog) also identify continental and regional differences in migration rates, which raises questions about appropriate scales of reconstruction/modelling that are relevant to the point about transferability.

Reply: Thank you, we agree and add both in the revised manuscript (L279-284):

“How strong postglacial migration lags are, is highly debated. The rate of migration depends on the species and varies in space and time. Many studies suggest no lags^{20,56} or relatively short lags of 1500 years at most^{26,57}. Others state that many plant species have not yet reached equilibrium with climate in the current interglacial¹⁸, particularly the Siberian larch forest⁴⁹ and European trees^{10,23}”

L237. Not implausible but these would be small populations which would make small pollen and seed contributions to biome reconstructions and migration. Migration rates may also have been influenced by wetness and landscape heterogeneity. Reflecting on how issues raised might influence uncertainty in both pollen and climate reconstruction/models would give a better sense of balance and integration, rather than the current organisation, which considers each method separately.

Reply: We thank the Reviewer for the suggestion of an alternative structure which would also be a meaningful way to present the discussion. Since the techniques of simulated and reconstructed biomes are very different, we decide to structure the discussion based on key topics. We keep the structure in the manuscript.

Regarding the comment on the tree refugia in East Asia, we fully agree that this statement need further clarification. We add this info according to the Reviewer's suggestion (L308-313):

"Thus, the establishment of trees early during the deglaciation is climatically not implausible. However, these populations would be small with low pollen and seed production. They may neither contribute to the migration of trees nor being detectable in the pollen records. Studies on macrofossils reveal that scattered populations of tree species can remain undetected by pollen data for several thousand years (cf. Jackson et al.¹⁷ and references therein)."

L250-4 offer a simplistic interpretation that is at odds with the range of uncertainties and potential complexities referred to in the preceding text. Please revise to offer a more nuanced interpretation, rather than privileging the model results.

Reply: We apologise if the impression has been given that we privilege the model results or consider the vegetation simulated in the model as "correct" and the reconstructions as "wrong". This was not our intention and does not reflect our opinion. We do not (yet) know the reasons for the mismatch. We do not rule out technical problems or other causes. These can have their origin in both the model and the reconstructions and are probably a combination of shortcomings in both methods. At the moment we are still at the beginning of the analyses on this, because many processes cannot be checked so far for technical reasons, since e.g. important processes (such as seed dispersal or soil dynamics) are not implemented in the models. But we think that the mismatch in forest development needs to be pointed out so that other groups can also research the causes.

We add a new sub-figure to the Figure 3, showing the change in openness-index in the reconstructions and the model results at all sites and at the chironomid-sites only. This further underlines the model-data mismatch.

Since this statement is formulated here as an assumption (i.e. that the forest follows the temperature), we leave the statement as it is at this point, but weaken it in the Conclusions by referring to other possible reasons (L342-345).

"The reasons for this discrepancy can be manifold. Several processes important for vegetation colonisation and migration are not implemented in the model. The

strongest mismatch occurs in Asia for which the lowest amount of data is available, leaving methodological origins of the model-data differences open.”

L261-2. You state that the model is consistent with pollen-independent reconstructions. However, the preceding text on this is brief and the language is highly qualitative (e.g. similar, weaker), rather than offering any quantitative comparison. To establish the model as robust and rigorous, it would be useful if the comparisons could be more quantitative (e.g. refer to BNS or fss) and take into account variations in the temporal/spatial coverage of evidence (e.g. sparse evidence for Siberia):

Reply: We agree, we have limited the evaluation to qualitative statements, as it basically is only important whether the climate in the model and in the reconstructions show a similar long-term trend and periods such as the Younger Dryas or the beginning of the Holocene occur synchronously. That is mostly the case. The averaged forest fraction on all continents agree relatively well at LGM and mid-Holocene, the mismatches occur in between. Thus, it is mostly important to check the spatial-temporal consistency of climatic trends and not necessarily the exact amount of the warming in terms of °C from the LGM to the Holocene. For a quantitative comparison, temporally-resolved (vegetation independent) terrestrial reconstructions would be needed, but there are only few of these records available.

To better underline the skill of the model, we have calculated the correlation coefficients between the time series in the model and in the pollen-independent climate reconstructions and we add a table with the results in the supplement (Supplementary Table 2). We furthermore add to the method part the following description (L463-475):

*“Most proxy records in our simulation validation are not available in temperature or precipitation units. Therefore, we quantify the similarity between our simulation, TraCE-21ka simulation, and the proxy records based on the similarity of the temporal patterns of the time series. We use Gaussian kernel correlation (GKC) which is a correlation-based similarity estimator for time series with irregular time axis⁷³. The correlations are mostly very high ($cor > 0.8$) showing the good agreement of the deglacial warming trends in our simulation and the proxy records (**Supplementary Tab. 2**). As the deglacial warming trend is the dominant temporal pattern in most time series, we additionally isolate millennial-scale temporal patterns (e.g. imprints of the Bølling/Allerød and Younger Dryas) using Gaussian smoothers. The millennial-scale correlations are lower than for the orbital trends but mostly significant and of comparable magnitude to the TraCE-21ka simulation.”*

L623, and assumptions involved in spatial generalisation/averaging L635). You refer to data availability, but do not mention the data quality; both are needed to demonstrate that the reconstruction and model are robust.

Reply: We agree and add information on the data quality in the method part:

L512-517: “In this study, the synthesis of biome reconstructions for the Northern Hemisphere (north of 30°N) prepared by Cao et al.²⁸ is used. It covers the last 40000 years and is based on fossil pollen records available in the Neotoma palaeoecology database (www.neotomadb.org). The pollen dataset has been

taxonomically harmonized and temporally standardized for each continent and has been tested to ensure the quality of the data^{28,75}. ”

L542-545: *“This may lead to a systematic misrepresentation of the area covered by forests in the reconstructions, although the Asian dataset is accepted as spatially and temporally representative of regional vegetation changes^{43,75}. ”*

L267-8. Much literature considers the issue of (dis)equilibrium but none is cited here. Please show your awareness of the existing literature and debates, beyond that of Svenning and colleagues, particularly relating to migrational lag. E.g. Williams et al. 2021 Nature Eco Evo.

Reply: Thank you for drawing our attention to the excellent paper by Williams et al. We insert a new paragraph in the Introduction to present a better overview on the exiting literature:

L76-95: “Most plant species have changed their geographical range since the LGM, some of them thousands of kilometres¹⁷. The rate of migration is species-dependent and varies in space and time. Studies on fossil pollen records reveal both, fast (= closely following the climate change) and slow (=lagging climate forcing) vegetation responses to the deglacial climate change¹⁸. For instance, some marine and terrestrial records reveal a vegetation dynamic in line with the millennial climate variability in parts of the northern extratropics (excluding the Asian continent)¹⁹, in particular along the North Atlantic²⁰. Fast responses have also been reported for some tree species in Europe²¹ and woody taxa in North America²². In contrast, most tree types in Europe are still responding to the deglacial climate signal and have not yet reached their potential geographical range²³. Similarly, the geographical range of major tree species in eastern North America is not in equilibrium with climate²⁴. Thus, it is not yet clear, how the forest biome in a hemispheric perspective has changed since the LGM or will change in response to future climate changes.

Climate is seen as main determinant of the geographical ranges at broad spatial scales²⁵. However many ecological processes act on longer time-scales than the rate of strong climate changes and slow-down the response to the climate forcing¹⁸. In particular, colonisation of previous unvegetated areas or bare rocks can be slow¹⁷. “

L271. The link from the lateglacial and Holocene to future is rather abrupt. The vegetation/species distribution starting point for the future is markedly different from the lateglacial (e.g. widespread distribution of many species now, rather than lateglacial refugial starting point) and, as above, there is an extensive literature on the ability of trees to track climate change (past and future) which is not referenced. Please review the conclusions to ensure they are better supported by the temporal focus and findings of the study.

Reply: We fully agree. The focus of the study is the past and not the future. However, we think it is useful to give a small outlook on the future based on the results for the past. Therefore, we would like to keep this part in the manuscript. Since the reviewer is right about the different initial states, we extend our text by adding further references and by focusing more on the model to better fit the rest of the paper. This part now reads (L356-372):

“Although past vegetation distributions with open land-scapes and trees in refugia substantially differ from today’s situation with widespread species and massive human intervention, the results of this study may point to a problem in the ongoing and future anthropogenic climate change debate. Within the next 100-150 years, vegetation is projected to face a global mean temperature rise comparable to the magnitude of the warming during the deglaciation, but on much shorter time-scales⁷. Though the rate of vegetation change is increasing since the late Holocene⁶¹, the change in plant distributions is expected to lag this rapid climate change^{8,9}, leading to strong structural and compositional transformations and consequences for the functioning of ecosystems⁶². Our study suggests that the NH forest biome in its entirety may respond much slower to current and future ongoing climatic changes than Earth System Models which lack important processes to represent these dynamics, project. This may result in false projections of the vegetation cover and misrepresentations of the terrestrial carbon storage and vegetation-atmosphere interactions.”

L521. How do you define ‘natural vegetation’?

Reply: With “natural vegetation” we define in the model those plant functional types that have not been grown by humans. In contrast, land use can be prescribed as forcing in the model, but it has not been prescribed in the simulation described here. We specified this the first time we mention it in the revised manuscript (Method part L383)

Extended data figure 2. Please add what size time steps are used in (a).

Reply: The temporal and spatial coverage of the records is displayed in form of **(a)** the number of time steps available for each site and **(b)** the number of available sites at each time-step, for the entire region (NH > 30°N, total) and all individual continents. The entire analysis period covers 35 potential time steps. During the Holocene, temporal resolution is 500 years. During the deglaciation, temporal resolution is 1000 years (cf. Cao et al., 2019a).

Extended data figure 4. There are numerous low similarity reconstruction/simulations in the east central area of North America: is this in keeping with previous studies (which are numerous)? This better-studied area seems an ideal location to consider some of the potential uncertainties and/or unknowns, for comparison with the more data-deficient parts of Asia that form the focus of the current text. Giving a comparison would allow the paper to present a stronger argument.

Reply: We fully agree. The simulated biomes in east central area of North America reveal similarly low similarity as in Asia. This is related to the fact, that the model simulates too low tree cover there to be assigned to the forest biome. At the temperate forest-steppe transition zone, the model results suggest a broader extent of the forests, whereas the reconstructions show steppe vegetation, pointing to a (false?) simulation of the forest margin. We assume that this is (similar to the Asian monsoon margin) related to the overestimated moisture availability in the model in that region. The low similarity in the east central area of North America - thus - results from both, the absence of forests in the simulation in the more central parts and the different interpretation of the forest margin. Both are systematically during the simulation. It is not comparable

with the Asian vegetation, but we agree, that it will be an ideal test-bed to clarify potential causes for the mismatch in future research.

Reviewer #3 (Remarks to the Author):

This manuscript presents a data-model comparison of vegetation through the LGM to present, with a focus on mis-matches between simulated vegetation via a dynamic vegetation model (JSBACH), and reconstructed vegetation, classified from pollen records (biomisation method), across the northern hemisphere (north of 30 degrees). The noteworthy result is that the evolution of JSBACH-modelled tree cover does not match biomised pollen through the LGM. The authors explain that there are really two plausible reasons why this is the case: either biomisation isn't very good, or vegetation and soil dynamics are poorly captured in ESMs.

I am not surprised by these results, and I would expect that the majority of scientists who are familiar with ecological processes would expect this result. The implication is that pollen data are not ideal for estimating climate i.e. the results of studies such as Mauri et al 2015 should be viewed with extreme caution as vegetation is not in equilibrium with climate. I feel that we have known this already for at least a decade, but perhaps this point has not been sufficiently well made in the literature. The implication is not drawn out in this study, though it is an important point to make as these data are still perhaps uncritically used.

Reply: We thank Reviewer 3 for his/her constructive comments to our manuscript. We fully agree. This is exactly what we intend to say. The pollen-based reconstructions should be used with cautions as long as we cannot rule out the possibility that the multi-millennial disequilibrium is real and the conundrum is not only a technical problem. However, vegetation also consist of other plants, not only tree species and thus we do not know, to what extent the pollen-based climate reconstructions are affected and to what extent it is a problem of the pollen records or the vegetation (see e.g. the temporal mismatch between macrofossils and pollen reconstructions and climate reconstructions of the two). We will prove both in future research.

The manuscript offers a series of problems and plausible explanations of the discrepancy, but does not offer solutions to overcome these problems.

Reply: The aim of the paper is to point at the problem and to introduce possible explanations. We see this paper as motivation for other scientists to join working on it. Unfortunately, we cannot yet attribute the mismatch to specific processes nor can we really offer solutions, since the impact of too many potentially important processes cannot be quantified at the moment (they need new model developments). But if the lag in vegetation response to climate is real, it would have far reaching consequences as - like it is said by Reviewer #3 - pollen-based climate reconstructions are uncritically used for model evaluation and comparisons, also for time-slice simulations. Therefore, we decided to publish the results now.

Taking first the suggestion that biomisation may not represent vegetation. A possible solution is to apply alternative methods to the pollen data. The state-of-the-art now appears to be quantitative reconstruction via methods such as

REVEALS, and full Holocene reconstructions are now available for different parts of Eurasia at least (e.g. Li et al 2020; Githumbi et al 2022). These have the advantage of estimating forest fraction in grid cells, rather than the classified biomisation approach.

Reply: We fully agree and we currently work on a comparison of the REVEALS-based estimates and the model results for Europe and China for the Holocene. This has the advantage, that we can compare directly on plant-functional type (PFT) level. However, as these REVEALS-based reconstructions and the model do not share the same PFTs and REVEALS simulates only the vegetation composition whereas the model also considers the area of the grid-cells that are not covered by vegetation, the comparison is not that straightforward either. A northern hemispheric dataset is not available yet and available datasets mainly cover the Holocene, but not the deglaciation. REVEALS will definitely be an option to further challenge the conundrum. We thus include in the text (L232-239):

“One possibility to overcome the problems of the biomisation methods would be the direct comparison of the simulated tree-PFTs with pollen-based, quantitative estimates of tree coverage such as provided by the REVEALS-model⁴⁴. In this respect, much progress has been made in recent years and data sets, mostly for the last 10,000 years, have been published^{41,45}. However, a northern hemispheric dataset for the deglaciation is not available yet and some problems, e.g. regarding the temporal changes in pollen productivity, are also affecting the REVEALS results.”

There remain problems of parameterising these models if pollen production varies during periods of major climate upheaval, but this is a problem inherent in any analysis of pollen data. The authors might reflect on this as a direction of travel to overcome the problem that is faced (or at least a way of testing whether it is the biomisation process itself that causes the mis-match).

Reply: This is another shortcoming of the REVEALS model, the pollen productivity is constant with time, but nevertheless it is worth comparing the model results with the REVEALS output. According to the Reviewer’s suggestion we include this opportunity (see last comment)

I agree entirely with the authors that vegetation dynamics are not well captured in the ESM. But the authors do not suggest a solution, only point out the problem.

Reply: It is not easy to offer solutions, because we do not know which processes contribute to the mismatch. We can only point out the problems, but so far we cannot test the effect of these different factors such as permafrost or the absence of dynamic soils on the vegetation. These components have to be implemented and tested in the models, but it would be a major effort, i.e. a new project, to do it. Therefore, we unfortunately cannot come up with other solutions as recommending to implement the missing processes and to work on shortcomings in the reconstruction methods and to test if they change the results.

In my view what is needed is a thorough review of possible rates of migration of different tree species (note: not all trees are the same), placed within a "velocity of climate change" framework. The climate simulations should enable us to

calculate velocities of climate change (which vary across time and space, and are strongly controlled by eg. topography), and comparison can be made with the possible velocities of plant migration (for different tree species). This would enable us to assess whether trees kept up with, or significantly lagged, climate in the past, as a lesson for the present and future.

Reply: This is an excellent idea, but we doubt whether it is really possible. The model simulation was carried out with a coarse spatial resolution and one cannot expect the subgrid-scale vegetation and climate dynamics to be accurately reproduced. The orography is smoothed in the model, which can lead to differences between the climate in the model and in reality in mountainous regions. Apart from that, the migration rate is influenced by many non-climatic factors that are not included in the model. The question is how much would be gained over studies comparing pollen-independent climate reconstructions with vegetation dynamics and estimating migration rates.

This point is important in the implications of the study, that trees may not track future climate. Line 271 implies that current climatic shifts will be stronger than during deglaciation. I strongly question this statement: the authors declare a 5-7K shift across the LGM, which is understood to have taken place in < 50 years (Steffenson et al 2008). Current disruptions to the climate system are not on that order of magnitude. I agree that future change is alarming and of major societal concern, but the statement needs refining.

Reply: We agree that this statements was not precise enough. We here refer to the hemispheric and global mean climate change. The magnitude of 5-7 K is comparable with results of business-as-usual scenarios for the future, without efforts to reduce CO2 emissions. We change it to (L359-362):

“Within the next 100-150 years, vegetation is projected to face a global mean temperature rise comparable to the magnitude of the warming during the deglaciation, but on much shorter time-scales⁷.“

A wider question that should be addressed is the implication for land cover feedbacks in ESMs. If DVMs are a component of the ESMs and feedback to force the climate system, then if these are not correct then that has major implications for the validity of results of ESMs. Work is now showing that land cover plays an important forcing role in the climate system (Strandberg et al 2022). The authors may want to reflect this in their discussions.

Reply: We fully agree that land-cover feedback can substantially alter the climate. We add to the discussion (L288-291):

“ Mismatches in simulated forest cover may be amplified by vegetation-atmosphere interactions that may contribute to the more rapid forest expansion in the model compared to the reconstructions, at least on regional level.“

The methodologies that are used in the paper are sound, and descriptions are adequate to reproduce the work.

Reply: Thank you.

Mauri, A., Davis, B. A. S., Collins, P. M., & Kaplan, J. O. (2015). The climate of Europe during the Holocene: a gridded pollen-based reconstruction and its multi-proxy evaluation. *Quaternary Science Reviews*, 112, 109-127.

Strandberg, G., Lindström, J., Poska, A., Zhang, Q., Fyfe, R., Githumbi, E., ... & Gaillard, M. J. (2022). Mid-Holocene European climate revisited: New high-resolution regional climate model simulations using pollen-based land-cover. *Quaternary Science Reviews*, 281, 107431.

Li, F., Gaillard, M.-J., Cao, X., Herzschuh, U., Sugita, S., ..(18).. and Jia, X. (2020). Towards quantification of Holocene anthropogenic land-cover change in temperate China: A review in the light of pollen-based REVEALS reconstructions of regional plant cover. *Earth-Science Reviews*, 203:103119

Steffensen, J. P., Andersen, K. K., Bigler, M., Clausen, H. B., Dahl-Jensen, D., Fischer, H., ... & White, J. W. (2008). High-resolution Greenland ice core data show abrupt climate change happens in few years. *science*, 321(5889), 680-684.

Strandberg, G., Lindström, J., Poska, A., Zhang, Q., Fyfe, R., Githumbi, E., ... & Gaillard, M. J. (2022). Mid-Holocene European climate revisited: New high-resolution regional climate model simulations using pollen-based land-cover. *Quaternary Science Reviews*, 281, 107431.

References (copied from the manuscript, the newly added ones are marked in red):

1. Birks, H. J. B. Strengths and Weaknesses of Quantitative Climate Reconstructions Based on Late-Quaternary Biological Proxies. *Open Ecol. J.* **3**, 68–110 (2011).
2. Chevalier, M. *et al.* Pollen-based climate reconstruction techniques for late Quaternary studies. *Earth-Science Reviews* vol. 210 103384 (2020).
3. Bartlein, P. J. *et al.* Pollen-based continental climate reconstructions at 6 and 21 ka: A global synthesis. *Clim. Dyn.* **37**, 775–802 (2011).
4. Brierley, C. M. *et al.* Large-scale features and evaluation of the PMIP4-CMIP6 midHolocene simulations. **16**, 1847–1872 (2020).
5. Kageyama, M. *et al.* The PMIP4 Last Glacial Maximum experiments: Preliminary results and comparison with the PMIP3 simulations. *Clim. Past* **17**, 1065–1089 (2021).
6. Harrison, S. BIOME 6000 DB classified plotfile version 1. (2017) doi:10.17864/1947.99.
7. Loarie, S. R. *et al.* The velocity of climate change. *Nature* **462**, 1052–1055 (2009).
8. Svenning, J. C. & Sandel, B. Disequilibrium vegetation dynamics under future climate change. *American Journal of Botany* vol. 100 1266–1286 (2013).
9. Neilson, R. P. *et al.* *Forecasting Regional to Global Plant Migration in Response to Climate Change.* *BioScience* vol. 55 <https://academic.oup.com/bioscience/article/55/9/749/285963> (2005).
10. Normand, S. *et al.* Postglacial migration supplements climate in determining plant species ranges in Europe. *Proc. R. Soc. B Biol. Sci.* **278**, 3644–3653 (2011).
11. Seltzer, A. M. *et al.* Widespread six degrees Celsius cooling on land during the Last Glacial Maximum. **593**, (2021).
12. Tierney, J. E. *et al.* Glacial cooling and climate sensitivity revisited. *Nature* **584**, 569 (2020).

13. Ray, N. & Adams, J. M. A GIS-based Vegetation Map of the World at the Last Glacial Maximum (25,000-15,000 BP). *Internet Archaeol.* (2001) doi:10.11141/ia.11.2.
14. Birks, H. J. B. & Willis, K. J. Alpines, trees, and refugia in Europe. *Plant Ecol. Divers.* **1**, 147–160 (2008).
15. Roberts, D. R. & Hamann, A. Glacial refugia and modern genetic diversity of 22 western North American tree species. *Proc. R. Soc. B Biol. Sci.* **282**, (2015).
16. Clark, J. S. Why trees migrate so fast: Confronting theory with dispersal biology and the paleorecord. *Am. Nat.* **152**, 204–224 (1998).
17. Jackson, S. T., Overpeck & Jonathan T. Responses of plant populations and communities to environmental changes of the late Quaternary. (2000) doi:10.1666/0094.
18. Williams, J. W., Ordonez, A. & Svenning, J.-C. A unifying framework for studying and managing climate-driven rates of ecological change. *Nat. Ecol. Evol.* **5**, (2021).
19. Harrison, S. P. & Goñi, M. F. S. Global patterns of vegetation response to millennial-scale variability and rapid climate change during the last glacial period. *Quat. Sci. Rev.* **29**, 2957–2980 (2010).
20. Williams, J. W., Post, D. M., Cwynar, L. C., Lotter, A. F. & Levesque, A. J. Rapid and widespread vegetation responses to past climate change in the North Atlantic region. *Geology* **30**, 971–974 (2002).
21. Giesecke, T., Brewer, S., Finsinger, W., Leydet, M. & Bradshaw, R. H. W. Patterns and dynamics of European vegetation change over the last 15,000 years. *J. Biogeogr.* **44**, 1441–1456 (2017).
22. Ordonez, A. & Williams, J. W. Climatic and biotic velocities for woody taxa distributions over the last 16 000 years in eastern North America. *Ecol. Lett.* **16**, 773–781 (2013).
23. Svenning, J.-C. & Skov, F. Limited filling of the potential range in European tree species. *Ecol. Lett.* **7**, 565–573 (2004).
24. Talluto, M. V., Boulangeat, I., Vissault, S., Thuiller, W. & Gravel, D. Extinction debt and colonization credit delay range shifts of eastern North American trees. *Nat. Ecol. Evol.* **1**, 1–6 (2017).
25. Pearson, R. G. & Dawson, T. P. Predicting the impacts of climate change on the distribution of species: are bioclimate envelope models useful? *Glob. Ecol. Biogeogr.* **12**, 361–371 (2003).
26. Webb, T. Is vegetation in equilibrium with climate? How to interpret late-Quaternary pollen data. *Vegetatio* **67**, 75–91 (1986).
27. Jackson, S. T. & Williams, J. W. Modern analogs in quaternary paleoecology: Here today, gone yesterday, gone tomorrow? *Annual Review of Earth and Planetary Sciences* vol. 32 495–537 (2004).
28. Cao, X., Tian, F., Dallmeyer, A. & Herzschuh, U. Northern Hemisphere biome changes (>30°N) since 40 cal ka BP and their driving factors inferred from model-data comparisons. *Quat. Sci. Rev.* **220**, 291–309 (2019).
29. He, F. *SIMULATING TRANSIENT CLIMATE EVOLUTION OF THE LAST DEGLACIATION WITH CCSM3*. (2011).
30. Osman, M. B. *et al.* Globally resolved surface temperatures since the Last Glacial Maximum. *Nature* **599**, 239–244 (2021).
31. Shakun, J. D. *et al.* Global warming preceded by increasing carbon dioxide concentrations during the last deglaciation. *Nature* **484**, 49–54 (2012).
32. Alley, R. B. The Younger Dryas cold interval as viewed from central Greenland. in *Quaternary Science Reviews* vol. 19 213–226 (Pergamon, 2000).

33. He, C. *et al.* Hydroclimate footprint of pan-Asian monsoon water isotope during the last deglaciation. *Sci. Adv.* **7**, (2021).
34. Reick, C. H., Raddatz, T., Brovkin, V. & Gayler, V. Representation of natural and anthropogenic land cover change in MPI-ESM. **5**, 1–24 (2013).
35. Prentice, I. C., Guiot, J., Huntley, B., Jolly, D. & Cheddadi, R. Reconstructing biomes from palaeoecological data: A general method and its application to European pollen data at 0 and 6 ka. *Clim. Dyn.* **12**, 185–194 (1996).
36. Dallmeyer, A., Claussen, M. & Brovkin, V. Harmonising plant functional type distributions for evaluating Earth system models. *Clim. Past* **15**, 335–366 (2019).
37. Ni, J., Cao, X., Jeltsch, F. & Herzschuh, U. Biome distribution over the last 22,000 yr in China. *Palaeogeogr. Palaeoclimatol. Palaeoecol.* **409**, 33–47 (2014).
38. Williams, J. W. & Jackson, S. T. Novel climates, no-analog communities, and ecological surprises. *Frontiers in Ecology and the Environment* vol. 5 475–482 (2007).
39. Sobol, M. K., Scott, L. & Finkelstein, S. A. Reconstructing past biomes states using machine learning and modern pollen assemblages: A case study from Southern Africa. *Quat. Sci. Rev.* **212**, 1–17 (2019).
40. Marinova, E. *et al.* Pollen derived biomes in the Eastern Mediterranean–Black Sea–Caspian Corridor. *J. Biogeogr.* **45**, 484–499 (2018).
41. Cao, X. *et al.* Pollen-based quantitative land-cover reconstruction for northern Asia covering the last 40 ka cal BP. *Clim. Past* **15**, 1503–1536 (2019).
42. Geng, R. *et al.* Modern Pollen Assemblages From Lake Sediments and Soil in East Siberia and Relative Pollen Productivity Estimates for Major Taxa. *Front. Ecol. Evol.* **10**, 508 (2022).
43. Cao, X. *et al.* A taxonomically harmonized and temporally standardized fossil pollen dataset from Siberia covering the last 40 kyr. *Earth Syst. Sci. Data* **12**, 119–135 (2020).
44. Sugita, S. Theory of quantitative reconstruction of vegetation I: pollen from large sites REVEALS regional vegetation composition. *The Holocene* **17**, 229–241 (2007).
45. Githumbi, E. *et al.* European pollen-based REVEALS land-cover reconstructions for the Holocene: Methodology, mapping and potentials. *Earth Syst. Sci. Data* **14**, 1581–1619 (2022).
46. Snell, R. S. *et al.* Using dynamic vegetation models to simulate plant range shifts. *Ecography (Cop.)*. **37**, 1184–1197 (2014).
47. Bullock, J. M. *et al.* Modelling spread of British wind-dispersed plants under future wind speeds in a changing climate. *J. Ecol.* **100**, 104–115 (2012).
48. Svenning, J. C., Normand, S. & Skov, F. Postglacial dispersal limitation of widespread forest plant species in nemoral Europe. *Ecography (Cop.)*. **31**, 316–326 (2008).
49. Herzschuh, U. *et al.* Glacial legacies on interglacial vegetation at the Pliocene-Pleistocene transition in NE Asia. *Nat. Commun.* **7**, 1–11 (2016).
50. Herzschuh, U. Legacy of the Last Glacial on the present day distribution of deciduous versus evergreen boreal forests. *Glob. Ecol. Biogeogr.* **29**, 198–206 (2020).
51. Väliranta, M. *et al.* Plant macrofossil evidence for an early onset of the Holocene summer thermal maximum in northernmost Europe. *Nat. Commun.* **6**, 1–8 (2015).
52. Davis, M. B., Shaw, R. G. & Etterson, J. R. EVOLUTIONARY RESPONSES TO CHANGING CLIMATE. *Ecology* **86**, 1704–1714 (2005).

53. Urban, M. C., Tewksbury, J. J. & Sheldon, K. S. On a collision course: Competition and dispersal differences create no-analogue communities and cause extinctions during climate change. *Proc. R. Soc. B Biol. Sci.* **279**, 2072–2080 (2012).
54. Pennington, W. Lags in adjustment of vegetation to climate caused by the pace of soil development. Evidence from Britain. *Vegetatio* **67**, 105–118 (1986).
55. MacDonald, G. M., Kremenetski, K. V. & Beilman, D. W. Climate change and the northern Russian treeline zone. *Philosophical Transactions of the Royal Society B: Biological Sciences* vol. 363 2285–2299 (2008).
56. Harrison, S. P. & Sanchez Goñi, M. F. Global patterns of vegetation response to millennial-scale variability and rapid climate change during the last glacial period. doi:10.1016/j.quascirev.2010.07.016.
57. Prentice, I. C., Bartlein, P. J. & Webb, T. Vegetation and climate change in eastern North America since the last glacial maximum. *Ecology* **72**, 2038–2056 (1991).
58. Cao, X. yong, Herzschuh, U., Telford, R. J. & Ni, J. A modern pollen-climate dataset from China and Mongolia: Assessing its potential for climate reconstruction. *Rev. Palaeobot. Palynol.* **211**, 87–96 (2014).
59. Leroy, S. A. G., Arpe, K., Mikolajewicz, U. & Wu, J. Climate simulations and pollen data reveal the distribution and connectivity of temperate tree populations in eastern Asia during the Last Glacial Maximum. *Clim. Past* **16**, 2039–2054 (2020).
60. Kaufman, D. *et al.* A global database of Holocene paleotemperature records. *Sci. Data* **7**, (2020).
61. Mottl, O. *et al.* Global acceleration in rates of vegetation change over the past 18,000 years. *Science* vol. 372 860–864 (2021).
62. Nolan, C. *et al.* Past and future global transformation of terrestrial ecosystems under climate change. *Science (80-.)*. **361**, 920–923 (2018).
63. Mauritsen, T. *et al.* Developments in the MPI-M Earth System Model version 1.2 (MPI-ESM1.2) and Its Response to Increasing CO₂. *J. Adv. Model. Earth Syst.* **11**, 998–1038 (2019).
64. Reick, C. *et al.* JSBACH 3 - The land component of the MPI Earth System Model: documentation of version 3.2. Hamburg: MPI für Meteorologie. *Berichte zur Erdsystemforsch.* (2021).
65. Brovkin, V., Raddatz, T., Reick, C. H., Claussen, M. & Gayler, V. Global biogeophysical interactions between forest and climate. *Geophys. Res. Lett.* **36**, L07405 (2009).
66. Berger, A. L. Long-term variations of daily insolation and Quaternary climatic changes. *Journal of Atmospheric Sciences* vol. 35 2361–2367 (1978).
67. Köhler, P., Nehrbass-Ahles, C., Schmitt, J., Stocker, T. F. & Fischer, H. A 156 kyr smoothed history of the atmospheric greenhouse gases CO₂, CH₄, and N₂O and their radiative forcing. *Earth Syst. Sci. Data* **9**, 363–387 (2017).
68. Tarasov, L., Dyke, A. S., Neal, R. M. & Peltier, W. R. A data-calibrated distribution of deglacial chronologies for the North American ice complex from glaciological modeling. *Earth Planet. Sci. Lett.* **315–316**, 30–40 (2012).
69. Loana Meccia, V. & Mikolajewicz, U. Interactive ocean bathymetry and coastlines for simulating the last deglaciation with the Max Planck Institute Earth System Model (MPI-ESM-v1.2). *Geosci. Model Dev.* **11**, 4677–4692 (2018).
70. Riddick, T., Brovkin, V., Hagemann, S. & Mikolajewicz, U. Dynamic hydrological discharge modelling for coupled climate model simulations of the last glacial cycle: The MPI-DynamicHD model version 3.0. *Geosci. Model Dev.* **11**, 4291–4316 (2018).

71. Kapsch, M., Mikolajewicz, U., Ziemer, F. & Schannwell, C. Ocean Response in Transient Simulations of the Last Deglaciation Dominated by Underlying Ice Sheet Reconstruction and Method of Meltwater Distribution. *Geophys. Res. Lett.* **49**, e2021GL096767 (2022).
72. Murton, J. B., Bateman, M. D., Dallimore, S. R., Teller, J. T. & Yang, Z. Identification of Younger Dryas outburst flood path from Lake Agassiz to the Arctic Ocean. *Nature* **464**, 740–743 (2010).
73. Rehfeld, K., Marwan, N., Heitzig, J. & Kurths, J. Comparison of correlation analysis techniques for irregularly sampled time series. *Nonlinear Process. Geophys.* **18**, 389–404 (2011).
74. Braconnot, P. *et al.* Evaluation of climate models using palaeoclimatic data. *Nature Climate Change* vol. 2 417–424 (2012).
75. Cao, X. Y., Ni, J., Herzschuh, U., Wang, Y. B. & Zhao, Y. A late Quaternary pollen dataset from eastern continental Asia for vegetation and climate reconstructions: Set up and evaluation. *Rev. Palaeobot. Palynol.* **194**, 21–37 (2013).
76. Bigelow, N. H. *et al.* Climate change and Arctic ecosystems: 1. Vegetation changes north of 55°N between the last glacial maximum, mid-Holocene, and present. *J. Geophys. Res. Atmos.* **108**, (2003).
77. Ramankutty, N. & Foley, J. A. Estimating historical changes in global land cover: Croplands from 1700 to 1992. *Global Biogeochem. Cycles* **13**, 997–1027 (1999).
78. Berger, A. & Loutre, M. F. Insolation values for the climate of the last 10 million years. *Quat. Sci. Rev.* **10**, 297–317 (1991).
79. Deplazes, G. *et al.* Links between tropical rainfall and North Atlantic climate during the last glacial period. *Nat. Geosci.* (2013) doi:10.1038/NCEO1712.

REVIEWERS' COMMENTS

Reviewer #1 (Remarks to the Author):

The authors have adequately answered to my questions, and the manuscript has been substantially improved regarding the concerns of the three reviewers. The only issue that the authors should be addressed is in paragraph 80. The authors say:

« For instance, some marine and terrestrial records reveal a vegetation dynamic in line with the millennial climate variability in parts of the northern extratropics (excluding the Asian continent)¹⁹»

This sentence is not clear. Why do the authors exclude the Asian continent? Because the lack of pollen records or due to the lack of vegetation response to the millennial-scale variability?

In the legend of Supplementary Figure 3, a) is the mean temperature of the coldest month while in the figure is indicated « Twarm ». The inversion is also observed for panel b) where « Tcold » is indicated for the mean temperature of the warmest month.

Once the authors clarify this issue and revised Supplementary Figure 3, I recommend the publication of the manuscript.

Reviewer #2 (Remarks to the Author):

The authors have responded clearly and in open-minded way to all reviewer comments. With regard to my comments, I am satisfied that they have made appropriate revisions. The reviewer comments and responses provide stimulating reading and the author revisions make this a stronger article overall, since it now gives a clearer indication of the uncertainties and complexities, and situates the work in a broader literature. Please see the annotated ms for a few specific comments and minor suggestions for editing, mainly on the revised text.

Reviewer #3 (Remarks to the Author):

I have read the author's responses to reviewers comments carefully, and cross-checked these with the revised manuscript. I am satisfied that the authors have addressed concerns and made positive change to the manuscript. Their purpose seems to be to establish, with this manuscript, a research agenda for the wider research community, and I think this is an appropriate direction to take.

Reply to REVIEWER COMMENTS: “The deglacial forest conundrum”

Reviewer #1 (Remarks to the Author):

The authors have adequately answered to my questions, and the manuscript has been substantially improved regarding the concerns of the three reviewers.

Reply: We would like to thank the reviewer again for his/her helpful comments on the manuscript. We revised our manuscript accordingly.

The only issue that the authors should be addressed is in paragraph 80. The authors say:

« For instance, some marine and terrestrial records reveal a vegetation dynamic in line with the millennial climate variability in parts of the northern extratropics (excluding the Asian continent)¹⁹”

This sentence is not clear. Why do the authors exclude the Asian continent? Because the lack of pollen records or due to the lack of vegetation response to the millennial-scale variability?

Reply: We apologize for any confusion. We wanted to point to the fact that the cited study is based on records outside the Asian continent, probably due to the lack of records. We change it to (L77-80):

“For instance, some marine and terrestrial records reveal a vegetation dynamic in line with the millennial climate variability in parts of the northern extratropics¹⁹, in particular along the North Atlantic²⁰. However, these studies do not include records from the Asian continent.”

In the legend of Supplementary Figure 3, a) is the mean temperature of the coldest month while in the figure is indicated « Twarm ». The inversion is also observed for panel b) where « Tcold » is indicated for the mean temperature of the warmest month.

Reply: Thank you for carefully reading the paper and the Supplement. The caption is indeed not correct (a and b are reversed). We change it.

Once the authors clarify this issue and revised Supplementary Figure 3, I recommend the publication of the manuscript.

Reply: Thank you, we change both.

Reviewer #2 (Remarks to the Author):

The authors have responded clearly and in open-minded way to all reviewer comments. With regard to my comments, I am satisfied that they have made appropriate revisions. The reviewer comments and responses provide stimulating reading and the author revisions make this a stronger article overall, since it now gives a clearer indication of the uncertainties and complexities, and situates the

work in a broader literature. Please see the annotated ms for a few specific comments and minor suggestions for editing, mainly on the revised text.

Reply: We would like to thank the reviewer for carefully reading the manuscript again and his/her thoughtful comments. We changed the language suggestions directly in the text (please see the document. We have switched on the “track changes” option). We have copied the specific comments of the referee into this reply (shown in black) and respond to them in detail (blue), in the following.

L88-89: Unclear: pollen does (with caveats) indicate how the forest biome (composition, distribution) has changed. I think you mean that we do not know how well it tracks climate at hemispheric scales

Reply: We fully agree, this was not well expressed. We change the sentence to (L86-89): “Thus, it is not yet clear, how well the forest biome in a hemispheric perspective tracked the climate changes since the LGM and how the hemispheric forest will respond to future climate changes.”

L93-94: Meaning is unclear. Do you mean lag behind climate forcing?

Reply: In principle yes. We changed the sentence to (L91-93): “However many ecological processes act on longer time-scales than the rate of strong climate changes and could thereby induce a lag in the vegetation response to the climate forcing”

L208-210: This text is too condensed to make a clear point. Please reconsider, e.g. highlight one or two biomes in fig 5S where the deviation may reflect these caveats

Reply: We apologize if this paragraph was too short and not precise enough. We change it to (L203-219): “The PFT-biomisation procedure is rather simple. It competes with other model biomisation techniques, but has – similar to these methods – difficulties to present biomes that are highly controlled by non-climatic parameters (such as the savanna) or multi-faceted biomes such as tundra³⁶ (Supplementary Fig. 5). Tundras can appear as a very open landscape with predominantly herbaceous vegetation, but can also contain a substantial proportion of small and shrubby trees (e.g. *Betula*, *Alnus* and *Salix*). Complicating this, taxa of tundra often have multiple life forms (e.g. shrub or tree) and a very wide ecological range. These problems make it challenging to clearly distinguish tundra from boreal forest, also in the reconstructions. With the limited number of PFTs in the vegetation models and the strict and simple definition of tundra in the simulated PFT-biomisation scheme, these (intra-biome) peculiarities cannot be taken into account. The differing definitions of the tundra biome in the model world and for the reconstructions may contribute to the mismatch in forest dynamic.”

L228-229: That may be the case now. It is also thought to have been rather homogeneous in the past?

Reply: Yes, the boreal forest in Siberia is mainly represented by Larch. The distribution of Larch is relatively homogeneous since the LGM (see the paper we

referred to by Cao et al, 2020). Also other studies based on pollen records and vegetation models suggest a relatively homogeneous distribution of the biome patterns since LGM at the continental scale (see e.g. Tian et al, 2018: <https://doi.org/10.1007/s00334-017-0653-8>). Of course the presence of cryptic vegetation, e.g. due to topography, permafrost, fire, etc. cannot be excluded at different sites.

L236-237: Not clear what this means. REVEALS analysis spanning NH deglaciation? Please be consistent in the use of abbreviations.

Reply: We agree, and write (L245-248): “However, a dataset covering the entire northern hemisphere and also the deglaciation period is not available yet and some problems, e.g. regarding temporal changes in pollen productivity through this period also affect the REVEALS results.”

L255: Meaning is unclear. misadaptation of trees in refugia to postglacial climate conditions?

Reply: Thank you, this was indeed misleading. It is discussed that tree populations in northern, glacial refugia have been highly adapted to the local climate and had a low genetic diversity due to poor pollen productivity, limited dispersal ability and relatively heavy seeds (e.g. *Picea* populations in Siberia). These genetically exhausted populations may have failed to expand during the deglaciation and may have lost the competition to other incoming populations that could expand more easily.

We change the sentence to (L264-268):

“For Siberia, lost competition strength of trees due to their adaptation to specific refugial conditions and low diversity is discussed⁵². These genetically exhausted populations may have failed to expand during the deglaciation. This would have necessitated a reinvasion of these taxa from southern populations.”

We added as reference: Schulte, L., Li, C., Livsovski, S. & Herzschuh, U. Forest-permafrost feedbacks and glacial refugia help explain the unequal distribution of larch across continents. *Journal of Biogeography* **9**, 0305-0270 (2022).

L291: Please add a citation

Reply: As this statement is rather general, we have not included a citation.

Reviewer #3 (Remarks to the Author):

I have read the author's responses to reviewers comments carefully, and cross-checked these with the revised manuscript. I am satisfied that the authors have addressed concerns and made positive change to the manuscript. Their purpose seems to be to establish, with this manuscript, a research agenda for the wider research community, and I think this is an appropriate direction to take.

Reply: Thank you very much for reviewing the manuscript a second time.